# Identifying Macro Causal Effects in a C-DMG over ADMGs

**Simon Ferreira**                                                                 *simon.ferreira@sorbonne-universite.fr*
*Sorbonne Université, INSERM, Institut Pierre Louis d'Epidémiologie et de Santé Publique,*
*F75012, Paris, France*

**Charles K. Assaad**                                                                 *charles.assaad@inserm.fr*
*Sorbonne Université, INSERM, Institut Pierre Louis d'Epidémiologie et de Santé Publique,*
*F75012, Paris, France*

**Reviewed on OpenReview:** *https://openreview.net/forum?id=905LEugq6R*

## Abstract

Causal effect identification using causal graphs is a fundamental challenge in causal inference. While extensive research has been conducted in this area, most existing methods assume the availability of fully specified directed acyclic graphs or acyclic directed mixed graphs. However, in complex domains such as medicine and epidemiology, complete causal knowledge is often unavailable, and only partial information about the system is accessible. This paper focuses on causal effect identification within partially specified causal graphs, with particular emphasis on cluster-directed mixed graphs (C-DMGs) which can represent many different acyclic directed mixed graphs (ADMGs). These graphs provide a higher-level representation of causal relationships by grouping variables into clusters, offering a more practical approach for handling complex systems. Unlike fully specified ADMGs, C-DMGs can contain cycles, which complicate their analysis and interpretation. Furthermore, their cluster-based nature introduces new challenges, as it gives rise to two distinct types of causal effects: macro causal effects and micro causal effects, each with different properties. In this work, we focus on macro causal effects, which describe the effects of entire clusters on other clusters. We establish that the *do-calculus* is both sound and complete for identifying these effects in C-DMGs over ADMGs when the cluster sizes are either unknown or of size greater than one. Additionally, we provide a graphical characterization of non-identifiability for macro causal effects in these graphs.

## 1 Introduction

A key challenge in causal inference for observational studies, known as the identification problem (Shpitser & Pearl, 2008), involves determining when and how a causal effect of a set of variables $\mathbb{X}$ on another set of variables $\mathbb{Y}$—denoted as $\Pr(\mathbb{Y} = \mathbb{y} \mid \operatorname{do}(\mathbb{X} = \mathbb{x}))$, where the $\operatorname{do}(\cdot)$ operator represents an external intervention—can be estimated from observational data, which are typically represented as a joint probability distribution under normal, intervention-free conditions. To address this problem, several graph-based techniques have been developed for causal effect identification, assuming a fully specified causal structure, often represented as directed acyclic graphs (DAGs) or acyclic directed mixed graphs (ADMGs). Among these methods, the do-calculus(Pearl, 1995) stands out as a sound and complete tool for causal effect identification. These tools were also extended to directed mixed graphs (DMGs) (Richardson, 1997; Forré & Mooij, 2017; 2018; Forré & Mooij, 2020; Boeken & Mooij, 2024).

However, constructing a fully specified causal graph is challenging because it requires knowledge of the causal relations among all pairs of observed variables. This knowledge is often unavailable, particularly in complex, high-dimensional settings, thus limiting the applicability of causal inference theory and tools. Therefore, there has recently been more interest in partially specified causal graphs (Maathuis & Colombo, 2013; Perkovic et al., 2016; Perkovic, 2020; Jaber et al., 2022; Wang et al., 2023; Anand et al., 2023; Wahl

et al., 2024; Boeken & Mooij, 2024; Assaad et al., 2023; 2024; Ferreira & Assaad, 2024; 2025). An important type of partially specified graphs is the Cluster-Directed Mixed Graph that can represent many ADMGs (C-DMG over ADMGs) which provides a coarser representation of causal relations through vertices that represent a cluster of variables (and can contain cycles). The partial specification of a C-DMG over ADMGs arises from the fact that each vertex represents a cluster of variables rather than a single variable. Within C-DMGs, two distinct types of causal effects exist: macro causal effects, which describe the causal influence between entire clusters, and micro causal effects, which capture the causal relationships between individual variables within and across clusters. In this paper, we focus on the former.

Recent work has demonstrated that the do-calculus can be directly applied to identify macro causal effects in certain restricted classes of C-DMG over ADMGs. Notably, these include Cluster-ADMGs, which enforce an acyclicity constraint on the cluster graph (Anand et al., 2023; Tikka et al., 2023), and summary causal graphs, where each cluster represents a single time series (Reiter et al., 2024; Ferreira & Assaad, 2025). In this study, we extend this line of work by addressing macro causal effect identification in general C-DMG over ADMGs, without imposing structural restrictions. Specifically, we allow for the presence of cycles and do not impose any constraints on how clusters are formed—any subset of variables can constitute a cluster as long as the cluster size is either unknown or it is of size greater than 1. More specifically, our contributions are as follows:

- We establish that d-separation (Pearl, 1988)—a fundamental tool in do-calculus—is sound and complete in C-DMG over ADMGs.

- We demonstrate that do-calculus remains sound and complete for identifying macro causal effects in C-DMG over ADMGs.

- We demonstrate that the classical concept of hedges (Shpitser & Pearl, 2006)—a graphical structure traditionally used to indicate the non-identifiability of causal effects—does not extend to C-DMG over ADMGs. Instead, we show that the recently introduced concept of SC-hedges (Ferreira & Assaad, 2025), which was originally developed for summary causal graphs (a specific subclass of C-DMG over ADMGs), is applicable to general C-DMG over ADMGs.

The remainder of the paper is organized as follows: In Section 2, we formally present C-DMG over AD-MGs and macro causal effects. In Section 3, we show that d-separation and the do-calculus is sound and complete for macro causal effects in C-DMG over ADMGs and present a graphical characterization for the non-identifiability of these effects. In Section 4, we examine the challenges that emerge when additional information about cluster sizes is available, particularly when some clusters have a size of 1. Finally in Section 5, we conclude the paper while showing its limitations. All proofs are deferred to the appendix.

## 2 Notation and Definitions

In this section, we introduce the key definitions and notations used throughout the paper to facilitate a clear and consistent presentation of our results. We begin by introducing the primary concept under consideration, namely the structural causal model (Pearl, 2009), also referred to as the probabilistic causal model (Shpitser & Pearl, 2008) which is assumed to be unknown.

**Definition 1** (Structural causal model (SCM) (Pearl, 2009))**.** *A structural causal model is a tuple* $\mathcal{M} = (\mathbb{L}, \mathbb{V}, \mathbb{F}, \Pr(\mathbb{L}))$*, where*

- $\mathbb{L}$ *is a set of exogenous variables, which cannot be observed but affect the rest of the model.*

- $\mathbb{V}$*, is a set of endogenous variables, which are observed and every* $V \in \mathbb{V}$ *is functionally dependent on some subset of* $\mathbb{L} \cup \mathbb{V} \setminus \{V\}$*.*

- $\mathbb{F}$ *is a set of functions such that for all* $V \in \mathbb{V}$*,* $f^V$ *is a function taking as input the values of a subset of* $\mathbb{L} \cup \mathbb{V} \setminus \{V\}$ *and outputting a value for* $V$*.*

- $\Pr(\mathbb{L})$ *is a joint probability distribution over* $\mathbb{L}$*.*

In this definition of SCMs, the set of functions $\mathbb{F}$ encapsulates the causal mechanisms governing the relationships among variables, while $\mathbb{L}$ represents the noise, *i.e.*, the unobserved or hidden variables that affect the observed variables, and $\mathbb{V}$ represents the observed variables, which are often referred to as *individual variables* or micro-variables or low-level variables. The uncertainty associated with these unobserved variables is characterized by the distribution $\Pr(\mathbb{l})$, which accounts for the unknown influences outside the observed system. When combined with the causal mechanisms encoded in $\mathbb{F}$, this distribution gives rise to the distribution $\Pr(\mathbb{v})$ over the observed variables $\mathbb{V}$. We also assume that each structural causal model (SCM) induces a directed acyclic graph (DAG), where every variable in $\mathbb{V} \cup \mathbb{L}$ corresponds to a vertex in the graph. In this DAG, a directed edge $\rightarrow$ is drawn from one variable to another if the former serves as an input to the function that determines the latter. For simplicity, instead of working directly with these DAGs, we consider an alternative representation known as an acyclic directed mixed graph (ADMG). In an ADMG, only the observed variables in $\mathbb{V}$ correspond to vertices, while hidden variables in $\mathbb{L}$ that share common inputs are represented by bidirected edges $\dashleftarrow\dashrightarrow$[1] between the corresponding observed variables, thereby implicitly accounting for the hidden confounding. Formally, ADMGs are defined as follows:

**Definition 2** (Acyclic directed mixed graph (ADMG)). *Consider an SCM $\mathcal{M}$. The ADMG $\mathcal{G} = (\mathbb{V}, \mathbb{E})$ induced by $\mathcal{M}$ is a graph where:*

- *the vertices $\mathbb{V}$ are the endogenous variables of the SCM; and*

- *$\forall X, Y \in \mathbb{V}$ the edge $X \rightarrow Y$ is in $\mathbb{E}$ if $Y$ is functionally dependent on $X$ and the edge $X \dashleftarrow\dashrightarrow Y$ is in $\mathbb{E}$ if both $X$ and $Y$ are functionally dependent on a same exogenous variable $L \in \mathbb{L}$.*

ADMGs have been extensively studied and are highly valuable in causal inference. However, in many fields such as epidemiology, constructing, analyzing, and validating an ADMG remains a significant challenge for researchers due to the inherent difficulty in accurately determining causal relationships among individual variables. This complexity primarily stems from the uncertainty surrounding causal relations, making it challenging to specify the precise structure of the graph. Nevertheless, researchers can often provide a partially specified version of the ADMG, which offers a more practical and compact representation of the underlying causal structure. These simplified representations, which we call Cluster-Directed Mixed Graphs (C-DMG over ADMGs), group several variables into clusters, allowing for the representation of causal relationships at a higher level of abstraction while retaining essential structural properties of the system. In a C-DMG, directed edges between clusters represent causal influences at the higher level, while bidirected edges capture hidden confounding effects that exist between clusters.

**Definition 3** (Cluster directed mixed graph over ADMGs (C-DMG over ADMGs)). *Let $\mathcal{G} = (\mathbb{V}, \mathbb{E})$ be an ADMG induced from an SCM $\mathcal{M}$. A C-DMG is a graph $\mathcal{G}^{\mathbb{c}} = (\mathbb{C}, \mathbb{E}^{\mathbb{c}})$ where:*

- *$\mathbb{C}$ is a partition (i.e., $k$ disjoints sets of micro-variables) of $\mathbb{V}$; and*

- *$\forall C_{\mathbb{X}}, C_{\mathbb{Y}} \in \mathbb{C}$ the edge $C_{\mathbb{X}} \rightarrow C_{\mathbb{Y}}$ (resp. $C_{\mathbb{X}} \dashleftarrow\dashrightarrow C_{\mathbb{Y}}$) is in $\mathbb{E}^{\mathbb{c}}$ if and only if there exists $X \in C_{\mathbb{X}}$ and $Y \in C_{\mathbb{Y}}$ such that $X \rightarrow Y$ (resp. $X \dashleftarrow\dashrightarrow Y$) is in $\mathbb{E}$.*

For simplicity, we will henceforth use the terms "C-DMG over ADMGs" and "C-DMGs" interchangeably. The transformation from an ADMG to a C-DMG can be made explicit using the natural transformation from $\mathbb{V}$ to $\mathbb{C} = \{C_{\mathbb{X}} = \{X_1, \cdots, X_{n^x}\}, \cdots, C_{\mathbb{Y}} = \{Y_1, \cdots, Y_{n^y}\}\}$:

$$(x_1, \cdots, x_{n^x}, \cdots, y_1, \cdots, y_{n^y}) \mapsto ((x_1, \cdots, x_{n^x}), \cdots, (y_1, \cdots, y_{n^y})).$$

The abstraction of C-DMG entails that, even though there is exactly one C-DMG compatible with a given ADMG, there are in general several ADMGs compatible with a given C-DMG. For example, we give in Figure 1 a very simple C-DMG with two of its compatible ADMGs. If all clusters in the C-DMG are of size equal to one, then the C-DMG according to our definition would be an ADMG. Notice that, a C-DMG

---

[1]In the literature, the bidirected edges in ADMGs are generally non-dashed. However, in this work, we use dashed bidirected edges to enhance visual clarity and distinction.

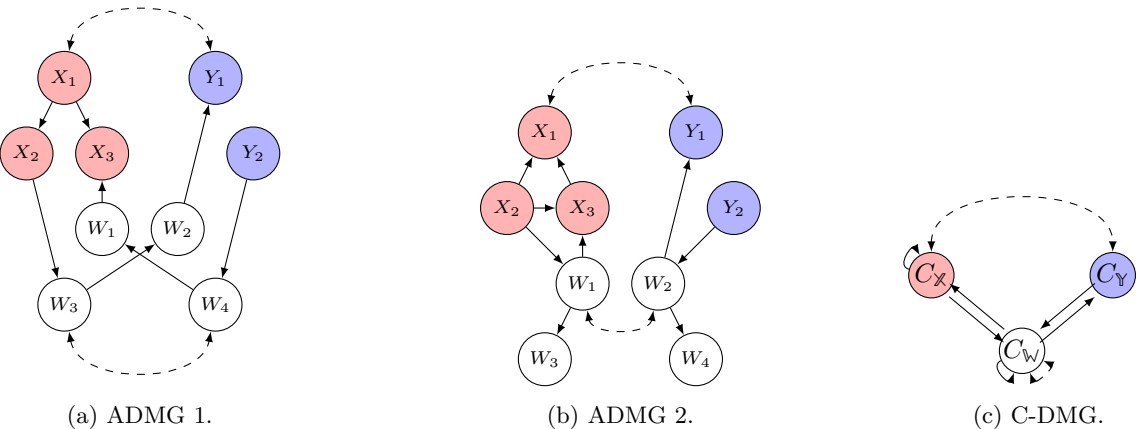

(a) ADMG 1.       (b) ADMG 2.      (c) C-DMG.

Figure 1: Two ADMGs and with their compatible C-DMG. Red vertices represent the exposures of interest in and blue vertices represent the outcome of interest.

may have directed cycles and in particular two directed edges oriented in opposite directions, however, we emphasize that all cycles in all C-DMGs that we consider in this paper, arise from the partial specificity. For example, in Figure 1 in the C-DMG, $C_\mathbb{X} \to C_\mathbb{W}$ and $C_\mathbb{W} \to C_\mathbb{X}$ form a cycle, which we often write $C_\mathbb{X} \rightleftarrows C_\mathbb{W}$. This cycle implies that in all ADMGs compatible with the C-DMG $\exists X_i, X_j \in C_\mathbb{X}$ and $W_k, W_l \in C_\mathbb{W}$ such that $X_i \to W_k$ and $X_j \leftarrow W_l$.

To streamline the presentation and avoid repetitive explanations of concepts applicable to both ADMGs and C-DMGs, we will adopt the unified notation $\mathcal{G}^* = (\mathbb{V}^*, \mathbb{E}^*)$ to refer to either type of graph. This notation allows us to generalize results and discussions without redundancy across different graph types. In the remainder, for every vertex $V^* \in \mathbb{V}^*$ in a graph $\mathcal{G}^* = (\mathbb{V}^*, \mathbb{E}^*)$ (whether it be an ADMG or a C-DMG), we will refer to its parents in graph by, $Pa(V^*, \mathcal{G}^*)$, its ancestors by $An(V^*, \mathcal{G}^*)$, and its descendants by $De(V^*, \mathcal{G}^*)$. We consider that a vertex counts as its own descendant and as its own ancestor. In addition, in a C-DMG $\mathcal{G}^c = (\mathbb{C}, \mathbb{E}^c)$, for every cluster $C_\mathbb{X} \in \mathbb{C}$, we will use the notation of strongly connected components defined as follows:

- **Strongly Connected Component:** $Scc(C_\mathbb{X}, \mathcal{G}^c) = An(C_\mathbb{X}, \mathcal{G}^c) \cap De(C_\mathbb{X}, \mathcal{G}^c)$.

Finally, in order to map the vertices in the C-DMG with the vertices in the ADMG, we will use the notion of corresponding cluster

- **Corresponding Cluster:** $\forall X \in C_\mathbb{X}, \; Cl(X, \mathcal{G}^c) = C_\mathbb{X}$.

We distinguish between two types of causal effects in the context of C-DMGs, the macro causal effect (Anand et al., 2023; Ferreira & Assaad, 2025) and the micro causal effect (Assaad et al., 2024; Assaad, 2025). In this paper we focus on the former and we formally define it below:

**Definition 4** (Macro causal effect). *Consider a SCM on variables $\mathbb{V}$ and a compatible C-DMG $\mathcal{G}^c = (\mathbb{C}, \mathbb{E}^c)$. A macro causal effect is a causal effect from a set of macro-variables $\mathbb{C}_\mathbb{X}$ on another set of macro-variables $\mathbb{C}_\mathbb{Y}$ where $\mathbb{C}_\mathbb{X}$ and $\mathbb{C}_\mathbb{Y}$ are disjoint subsets of $\mathbb{C}$. It is written $\Pr(\mathbb{C}_\mathbb{Y} = c_\mathbb{Y} \mid do(\mathbb{C}_\mathbb{X} = c_\mathbb{X}))$, where the $do(\cdot)$ operator represents an external intervention.*

In the following, we will abuse the notation by writing $\Pr(c_\mathbb{Y} \mid do(c_\mathbb{X}))$ instead of $\Pr(\mathbb{C}_\mathbb{Y} = c_\mathbb{Y} \mid do(\mathbb{C}_\mathbb{X} = c_\mathbb{X}))$ when the setting is clear.

The identification problem in causal inference aims to establish whether a causal effect of a set of variables on another set of variables can be expressed exclusively in terms of observed variables and standard probabilistic notions, such as conditional probabilities. Formally, the identification problem in the context of macro causal effects and C-DMGs is defined as follows:

**Definition 5** (Identifiability in C-DMGs)**.** *Let $\mathbb{X}$ and $\mathbb{Y}$ be disjoint sets of vertices in an unknown ADMG $\mathcal{G}$ compatible with a C-DMG $\mathcal{G}^c$. The causal effect of $\mathbb{X}$ on $\mathbb{Y}$ is identifiable in $\mathcal{G}^c$ if $\Pr(\mathbb{y} \mid do(\mathbb{x}))$ is uniquely computable from any observational positive distribution[2] compatible with $\mathcal{G}^c$.*

*Hence, there are no two ADMGs, $\mathcal{G}_1$ and $\mathcal{G}_2$ such that $\Pr_1(\mathbb{v}) = \Pr_2(\mathbb{v})$ where $\Pr_1$ is a positive distribution compatible with $\mathcal{G}_1$ and $\Pr_2$ is a positive distribution compatible with $\mathcal{G}_2$ and $\Pr_1(\mathbb{y} \mid do(\mathbb{x})) \neq \Pr_2(\mathbb{y} \mid do(\mathbb{x}))$.*

Whenever the ADMG is known, *i.e.*, all clusters in the C-DMG are of size equal to one, the *do-calculus* (Pearl, 1995) can be applied to identify the causal effect of interest. Recently, it has been shown that the do-calculus can also be applied when the ADMG is unknown but its C-DMG is known (with clusters of size greater than one) under one of the following conditions: the C-DMG is acyclic (Anand et al., 2023); or if the cluster corresponds to a time series (Ferreira & Assaad, 2025). However, it remains unclear whether the do-calculus can be directly applied to general C-DMGs. Therefore, the primary objective of this paper is to bridge this gap by developing analogous tools tailored for general C-DMGs.

We emphasize that, throughout this paper (except in Section 4), we operate under the following assumption:

**Assumption 1.** *The identifiability problem, as defined in Definition 5, is considered under one of the following assumptions:*

1. *The size of the clusters is unknown, or*

2. *No two adjacent clusters in a cycle are both of size one.*

While the first condition in the assumption may seem unrealistic—since, in practice, access to data would typically reveal cluster sizes—it remains relevant in early-stage study design. A modeler may wish to assess identifiability before finalizing which specific variables will be included in each cluster. For instance, an epidemiologist planning a study on the effect of salary on stress may know that multiple variables may represent salary, stress, and confounders, without yet specifying their exact composition. On the other hand, if the modeler already knows the precise variables in each cluster, and thus the cluster sizes, then our results apply only when there is no cycle containing two adjacent clusters both of size one (second condition in the assumption). In Section 4, we further analyze this assumption, distinguishing the results that remain valid when it is not satisfied from those that no longer hold.

## 3 Identification of Macro Causal Effects in a C-DMG over ADMGs

In this section, our primary objective is to demonstrate that the do-calculus is both sound and complete for identifying macro causal effects in C-DMGs. To achieve this, we first establish in the first subsection that d-separation, a fundamental tool used in do-calculus to determine conditional independencies, is also sound and complete within C-DMGs to determine "macro conditional independencies". In the second subsection, we present the main theoretical result of this section, providing a rigorous proof of the soundness and completeness of do-calculus for macro causal effect identification. Following this, we introduce a graphical characterization of non-identifiability, offering insights into scenarios where causal effects cannot be determined from observational data. Finally, in the last subsection, we explore the connection between our findings and summary causal graphs, demonstrating how our results extend and apply to this specific class of graphs.

### 3.1 The d-separation in a C-DMG over ADMGs

The standard definition of d-separation (Pearl, 1988) was introduced for ADMGs. It was later extended to DMGs in Forré & Mooij (2018) and to cluster-ADMGs in Anand et al. (2023). In this subsection, we show that it is also readily extendable to C-DMGs. We start by defining blocked paths and d-separation.

**Definition 6** (blocked walk (Pearl, 2009))**.** *In a graph $\mathcal{G}^* = (\mathbb{V}^*, \mathbb{E}^*)$ (whether it be an ADMG or a C-DMG), a walk $\tilde{\pi} = \langle V_1^*, \cdots, V_n^* \rangle$ is said to be blocked by a set of vertices $\mathbb{W}^* \subseteq \mathbb{V}^*$ if:*

---

[2]In this work, we call a positive distribution a distribution in which $\forall \mathbb{v}, \Pr(\mathbb{v}) > 0$, however this assumption can be loosened (Hwang et al., 2024).

1. $V_1^* \in \mathbb{W}^*$ or $V_n^* \in \mathbb{W}^*$, or

2. $\exists i : 1 < i < n$ such that $\langle V_{i-1}^* \leftrightarrowtail V_i^* \to V_{i+1}^* \rangle \subseteq \tilde{\pi}$ or $\langle V_{i-1}^* \leftarrow V_i^* \leftarrowtail V_{i+1}^* \rangle \subseteq \tilde{\pi}$ and $V_i^* \in \mathbb{W}^*$, or

3. $\exists i : 1 < i < n$ such that $\langle V_{i-1}^* \leftrightarrowtail V_i^* \leftarrowtail V_{i+1}^* \rangle \subseteq \tilde{\pi}$ and $De(V_i^*, \mathcal{G}^*) \cap \mathbb{W}^* = \varnothing$.

where $\leftrightarrowtail$ represents $\to$ or $\leftarrow\!\cdot\!\cdot\!\to$, $\leftrightarrow$ represents $\leftarrow$ or $\leftarrow\!\cdot\!\cdot\!\to$, and $\leftarrowtail$ represents any of the three arrow type $\to$, $\leftarrow$ or $\leftarrow\!\cdot\!\cdot\!\to$. A walk which is not blocked is said to be active.

**Definition 7** (d-separation (Pearl, 2009)). *In a graph $\mathcal{G}^* = (\mathbb{V}^*, \mathbb{E}^*)$ (whether it be an ADMG or a C-DMG), let $\mathbb{X}^*, \mathbb{Y}^*, \mathbb{W}^*$ be distinct subsets of $\mathbb{V}^*$. $\mathbb{W}^*$ is said to d-separate $\mathbb{X}^*$ and $\mathbb{Y}^*$ if and only if $\mathbb{W}^*$ blocks every path from a vertex in $\mathbb{X}^*$ to a vertex in $\mathbb{Y}^*$. It is written $(\mathbb{X}^* \perp\!\!\!\perp_d \mathbb{Y}^* \mid \mathbb{W}^*)_{\mathcal{G}^*}$.*

The following theorem shows that d-separation is applicable as is to C-DMGs.

**Theorem 1** (Soundness of d-separation in a C-DMG over ADMGs). *Let $\mathcal{G}^{\mathbb{c}} = (\mathbb{C}, \mathbb{E}^{\mathbb{c}})$ be a C-DMG and $\mathbb{C}_{\mathbb{X}}, \mathbb{C}_{\mathbb{Y}}, \mathbb{C}_{\mathbb{W}}$ be disjoint subsets of $\mathbb{C}$. If $\mathbb{C}_{\mathbb{X}}$ and $\mathbb{C}_{\mathbb{Y}}$ are d-separated by $\mathbb{C}_{\mathbb{W}}$ in $\mathcal{G}^{\mathbb{c}}$ then, in any compatible ADMG $\mathcal{G} = (\mathbb{V}, \mathbb{E})$, $\mathbb{X} = \bigcup_{C \in \mathbb{C}_{\mathbb{X}}} C$ and $\mathbb{Y} = \bigcup_{C \in \mathbb{C}_{\mathbb{Y}}} C$ are d-separated by $\mathbb{W} = \bigcup_{C \in \mathbb{C}_{\mathbb{W}}} C$.*

Theorem 1 shows that d-separation in C-DMGs guarantees finding some common macro-level d-separation in all compatible ADMGs which implies according to (Pearl, 2009, Theorem 1.2.5) that it allows the detection of some conditional independencies in the underlying probability distribution directly from the C-DMG. Thus, by extending the applicability of d-separation to C-DMGs, this result allows us to infer macro-level conditional independencies even when dealing with partially specified graphs. To appreciate the above result, we give in the following an example of the application of d-separation to C-DMGs

**Example 1.** *Let $\mathcal{G}$ be the true* unknown *ADMG and consider that its compatible C-DMG, denoted as $\mathcal{G}^{\mathbb{c}}$ is one given in Figure 2a. Using Definition 7, we can directly deduce $(C_{\mathbb{W}} \perp\!\!\!\perp_d C_{\mathbb{Y}} \mid C_{\mathbb{X}})_{\mathcal{G}^{\mathbb{c}}}$. Thus according to (Pearl, 2009, Theorem 1.2.5), $C_{\mathbb{W}}$ is conditionally independent of $C_{\mathbb{Y}}$ given $C_{\mathbb{X}}$ in every distribution compatible with the true ADMG.*

**Example 2.** *Let $\mathcal{G}$ be the true* unknown *ADMG and consider that its compatible C-DMG, denoted as $\mathcal{G}^{\mathbb{c}}$ is one given in in Figure 2c. Using Definition 7, we can directly deduce that $(C_{\mathbb{Z}} \perp\!\!\!\perp_d C_{\mathbb{Y}} \mid C_{\mathbb{X}}, C_{\mathbb{W}})_{\mathcal{G}^{\mathbb{c}}}$. Thus according to (Pearl, 2009, Theorem 1.2.5), $C_{\mathbb{Z}}$ is conditionally independent of $C_{\mathbb{Y}}$ given $C_{\mathbb{X}}$ and $C_{\mathbb{W}}$ in every distribution compatible with the true ADMG.*

The following theorem shows that d-separation is also complete in C-DMGs under Assumption 1.

**Theorem 2** (Completeness of d-separation in a C-DMG over ADMGs[3]). *Let $\mathcal{G}^c = (\mathbb{C}, \mathbb{E}^{\mathbb{c}})$ be a C-DMG, $\mathbb{C}_{\mathbb{X}}, \mathbb{C}_{\mathbb{Y}}, \mathbb{C}_{\mathbb{W}}$ be disjoint subsets of $\mathbb{C}$, $\mathbb{X} = \bigcup_{C \in \mathbb{C}_{\mathbb{X}}} C$, $\mathbb{Y} = \bigcup_{C \in \mathbb{C}_{\mathbb{Y}}} C$ and $\mathbb{W} = \bigcup_{C \in \mathbb{C}_{\mathbb{W}}} C$. Under Assumption 1, if $\mathbb{C}_{\mathbb{X}}$ and $\mathbb{C}_{\mathbb{Y}}$ are not d-separated by $\mathbb{C}_{\mathbb{W}}$ in $\mathcal{G}^{\mathbb{c}}$, then there exists a compatible ADMG $\mathcal{G} = (\mathbb{V}, \mathbb{E})$ such that $\mathbb{X}$ and $\mathbb{Y}$ are not d-separated by $\mathbb{W}$.*

The findings of the above theorem establish that identifying a d-separation in C-DMGs guarantees finding *all* common macro-level d-separation in all compatible ADMGs. This is particularly valuable in constraint-based causal discovery when the interest is to uncover the structure of the C-DMG without uncovering an ADMG. Most importantly, these findings serve as a critical foundation for the results presented in the next subsection.

## 3.2   The do-calculus in a C-DMG over ADMGs

The do-calculus initially introduced in Pearl (1995) is an important tool of causal inference that consists of three rules. It allows to express, whenever it is possible, queries under interventions, *i.e.*, those that contains a $\mathrm{do}(\cdot)$ operator, as queries that can be computed from positive observational distribution, *i.e.*, that does

---

[3]The proof of this Theorem follows the same intuition as the proof of Theorem 2 in Ferreira & Assaad (2025). However, Ferreira & Assaad (2025) made the implicit and unnecessary assumption that the clusters were of sizes bigger than the number of clusters. In our proof we do not need this assumption and thereby our proof generalize their result *i.e.*,the clusters do not need to be of size bigger than 2, nor for the completeness of d-separation in C-DMGs, nor for the completeness of d-separation in summary causal graphs.

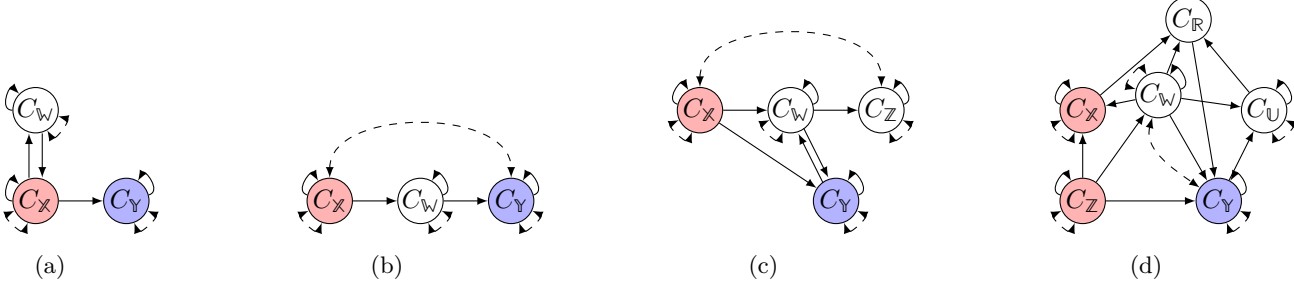

Figure 2: C-DMGs with identifiable macro causal effects. Each pair of red and blue vertices represents the causal effect we are interested in.

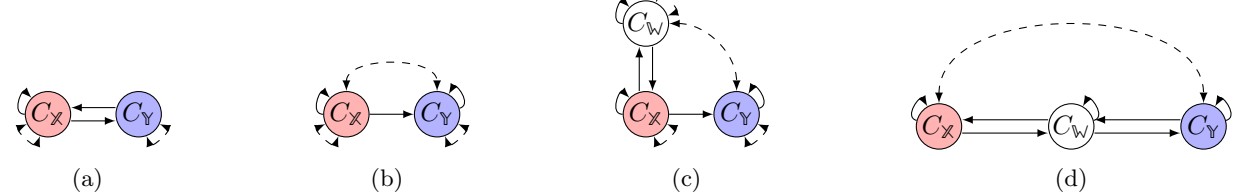

Figure 3: C-DMGs with not identifiable macro causal effects. Each pair of red and blue vertices represents the total effect we are interested in.

not contain a $do(\cdot)$ operator. In such cases, it is said that the query containing the $do(\cdot)$ is identifiable. The do-calculus was initially introduced for ADMGs so it is not easily extendable to cyclic graphs. In this subsection, we show that it is also readily extendable to C-DMGs (containing cycles).

Firstly, we define the notion of mutilated graphs (Pearl, 2009). Consider a causal graph $\mathcal{G}^* = (\mathbb{V}^*, \mathbb{E}^*)$ (whether it be an ADMG or a C-DMG) and $\mathbb{A}^*, \mathbb{B}^* \subseteq \mathbb{V}^*$, a mutilated graph denoted by $\mathcal{G}^*_{\overline{\mathbb{A}^*}\underline{\mathbb{B}^*}}$ is the graph obtained by removing all edges coming in $\mathbb{A}^*$ and all edges coming out of $\mathbb{B}^*$.

Using the notion of mutilated graphs and d-separation we show that the do-calculus is applicable to C-DMGs.

**Theorem 3** (do-calculus for a C-DMG over ADMGs and macro causal effects). *Let $\mathcal{G}^c = (\mathbb{C}, \mathbb{E}^c)$ be a C-DMG and $\mathbb{C}_{\mathbb{X}}, \mathbb{C}_{\mathbb{Y}}, \mathbb{C}_{\mathbb{Z}}, \mathbb{C}_{\mathbb{W}}$ be disjoint subsets of $\mathbb{C}$. The three following rules of the do-calculus are sound.*

$$\textbf{\textit{Rule 1:}}\Pr(\mathbb{c}_y \mid do(\mathbb{c}_z), \mathbb{c}_x, \mathbb{c}_w) = \Pr(\mathbb{c}_y \mid do(\mathbb{c}_z), \mathbb{c}_w) \qquad \textit{if } (\mathbb{C}_{\mathbb{Y}} \perp\!\!\!\perp_d \mathbb{C}_{\mathbb{X}} \mid \mathbb{C}_{\mathbb{Z}}, \mathbb{C}_{\mathbb{W}})_{\mathcal{G}^c_{\overline{\mathbb{C}_{\mathbb{Z}}}}}$$

$$\textbf{\textit{Rule 2:}}\Pr(\mathbb{c}_y \mid do(\mathbb{c}_z), do(\mathbb{c}_x), \mathbb{c}_w) = \Pr(\mathbb{c}_y \mid do(\mathbb{c}_z), \mathbb{c}_x, \mathbb{c}_w) \qquad \textit{if } (\mathbb{C}_{\mathbb{Y}} \perp\!\!\!\perp_d \mathbb{C}_{\mathbb{X}} \mid \mathbb{C}_{\mathbb{Z}}, \mathbb{C}_{\mathbb{W}})_{\mathcal{G}^c_{\overline{\mathbb{C}_{\mathbb{Z}}}\underline{\mathbb{C}_{\mathbb{X}}}}}$$

$$\textbf{\textit{Rule 3:}}\Pr(\mathbb{c}_y \mid do(\mathbb{c}_z), do(\mathbb{c}_x), \mathbb{c}_w) = \Pr(\mathbb{c}_y \mid do(\mathbb{c}_z), \mathbb{c}_w) \qquad \textit{if } (\mathbb{C}_{\mathbb{Y}} \perp\!\!\!\perp_d \mathbb{C}_{\mathbb{X}} \mid \mathbb{C}_{\mathbb{Z}}, \mathbb{C}_{\mathbb{W}})_{\mathcal{G}^c_{\overline{\mathbb{C}_{\mathbb{Z}}\mathbb{C}_{\mathbb{X}}(\mathbb{C}_{\mathbb{W}})}}}$$

*where $\mathbb{C}_{\mathbb{X}}(\mathbb{C}_{\mathbb{W}})$ is the set of vertices in $\mathbb{C}_{\mathbb{X}}$ that are non-ancestors of any vertex in $\mathbb{C}_{\mathbb{W}}$ in the mutilated graph $\mathcal{G}^c_{\overline{\mathbb{C}_{\mathbb{Z}}}}$.*

Using the soundness of the do-calculus in C-DMG (Theorem 3), we can easily use the rules of the do-calculus to find out that the causal effect $\Pr(\mathbb{c}_y \mid do(\mathbb{c}_x))$ is identifiable in all C-DMGs in Figure 2 as demonstrated in the following examples.

**Example 3.** *Both in Figure 2a and 2c, one can verify that $(C_{\mathbb{Y}} \perp\!\!\!\perp_d C_{\mathbb{X}})_{\mathcal{G}^c_{\underline{C_{\mathbb{X}}}}}$, thus Rule 2 of the do-calculus is applicable and $\Pr(c_y \mid do(c_x)) = \Pr(c_y \mid c_x)$.*

**Example 4.** *Notice that Figure 2b does not contain any cycle other than self-loops and is very similar to Figure 1(b) of Anand et al. (2023) which corresponds to the well-known front-door criterion (Pearl, 2009). Thus, using the corresponding sequence of classical rules of probability and rules of do-calculus as the one given in (Pearl, 2009, p.83), one obtains $\Pr(c_y \mid do(c_x)) = \sum_{c_w} \Pr(c_w \mid c_x) \sum_{c_{x'}} \Pr(c_y \mid c_w, c_{x'}) \Pr(c_{x'})$.*

**Example 5.** *Consider the C-DMG in Figure 2d containing a cycle between $c_\mathbb{Y}$, $c_\mathbb{R}$, and $c_\mathbb{U}$ and a hidden confounding between $c_\mathbb{Y}$ and $c_\mathbb{W}$. Let $\Pr(c_\mathbb{y} \mid do(c_\mathbb{x}, c_\mathbb{z}))$ be the causal effect of interest. Using the rule of total probability we can rewrite $\Pr(c_\mathbb{y} \mid do(c_\mathbb{x}, c_\mathbb{z}))$ as*

$$\sum_{c_\mathbb{w}} \Pr(c_\mathbb{y} \mid do(c_\mathbb{x}, c_\mathbb{z}), c_\mathbb{w})\Pr(c_\mathbb{w} \mid do(c_\mathbb{x}, c_\mathbb{z})).$$

*We first focus on $\Pr(c_\mathbb{y} \mid do(c_\mathbb{x}, c_\mathbb{z}), c_\mathbb{w})$. Notice that $(C_\mathbb{Y} \perp\!\!\!\perp_d C_\mathbb{X} \mid C_\mathbb{Z}, C_\mathbb{W})_{\mathcal{G}^c_{\overline{C_\mathbb{Z}} \underline{C_\mathbb{X}}}}$ and that $(C_\mathbb{Y} \perp\!\!\!\perp_d C_\mathbb{Z} \mid C_\mathbb{X}, C_\mathbb{W})_{\mathcal{G}^c_{\underline{C_\mathbb{Z}}}}$ which means by applying Rule 2 consecutively, the expression $\Pr(c_\mathbb{y} \mid do(c_\mathbb{x}, c_\mathbb{z}), c_\mathbb{w})$ can be equivalently rewritten as: $\Pr(c_\mathbb{y} \mid c_\mathbb{x}, c_\mathbb{z}, c_\mathbb{w})$.*

*Now we focus on $\Pr(c_\mathbb{w} \mid do(c_\mathbb{x}, c_\mathbb{z}))$. Notice that $(C_\mathbb{W} \perp\!\!\!\perp_d C_\mathbb{X} \mid C_\mathbb{Z})_{\mathcal{G}^c_{\overline{C_\mathbb{Z}} \overline{C_\mathbb{X}}}}$ which means by Rule 3 of the do-calculus we can completely remove $do(c_\mathbb{x})$ from the expression. Furthermore, we have $(C_\mathbb{W} \perp\!\!\!\perp_d C_\mathbb{Z})_{\mathcal{G}^c_{\underline{C_\mathbb{Z}}}}$ which means by Rule 2 we can replace $do(c_\mathbb{z})$ by $c_\mathbb{z}$. So we can rewrite $\Pr(c_\mathbb{w} \mid do(c_\mathbb{x}, c_\mathbb{z}))$ as $\Pr(c_\mathbb{w} \mid c_\mathbb{z})$.*

In these three examples, the rules of the do-calculus allow a macro causal effect to be expressed solely in terms of observed variables. Consequently, the macro causal effect can be estimated from data, provided that the positivity assumption holds.

In the following theorem, we show that the do-calculus is not only applicable to C-DMGs, but also it is complete.

**Theorem 4** (Completeness of do-calculus for a C-DMG over ADMGs and macro causal effects). *Under Assumption 1, if one of the do-calculus rules does not apply for a given C-DMG, then there exists a compatible ADMG for which the corresponding rule does not apply.*

Using the completeness of the do-calculus in C-DMGs (Theorem 4), we can determine that the causal effect $\Pr(c_\mathbb{y} \mid do(c_\mathbb{x}))$ is not identifiable in all C-DMGs depicted in Figure 3 and in the C-DMG in Figure 1 by examining all possible iterations of the rules of the do-calculus. However, it is well known that exhaustively examining all possibilities for applying the rules of do-calculus can quickly become impractical, particularly for large graphs. To address this challenge, the following subsection introduces a sub-graphical structure designed to directly determine whether it is feasible to express a macro causal effect solely in terms of observed variables using do-calculus and C-DMG.

Notice that, even though C-DMGs are defined with the inclusion of self directed edges (*e.g.*, $C \rightarrow C$) and self dashed bidirected edges (*e.g.*, $C \dashleftarrow\!\!\!\dashrightarrow C$), we only focused on paths in this paper and thus, under Assumption 1, adding or removing such edges will not influence identifiability results. However, this is not necessarily true if Assumption 1 is not verified. In addition, as shown in Assaad et al. (2024), self directed edges and self dashed bidirected edges are very useful in the identification of micro causal effects.

### 3.3 Non-Identifiability: a graphical characterization

In ADMGs, there exists a sub-graphical structure, called a hedge (Shpitser & Pearl, 2006), which is employed to graphically characterize non-identifiability as shown in Shpitser & Pearl, 2006, Theorem 4. To properly define it for C-DMGs, it is essential to first familiarize oneself with the two related definitions which we have adapted and provided below specifically for the context of C-DMGs:

**Definition 8** (C-component, Tian & Pearl (2002)). *Let $\mathcal{G}^* = (\mathbb{V}^*, \mathbb{E}^*)$ be a graph (whether it be a C-DMG or an ADMG). A subset of vertices $\mathbb{V}^*_C \subseteq \mathbb{V}^*$ such that $\forall V_1^*, V_n^* \in \mathbb{V}^*_C, \; \exists V_2^*, \cdots, V_{n-1}^* \in \mathbb{V}^*$ with $\forall 1 \le i < n, \; V_i^* \dashleftarrow\!\!\!\dashrightarrow V_{i+1}^*$ is called a C-component.*

**Definition 9** (C-forest, Shpitser & Pearl (2006)). *Let $\mathcal{G}^* = (\mathbb{V}^*, \mathbb{E}^*)$ be a graph (whether it be a C-DMG or an ADMG). If $\mathcal{G}^*$ is acyclic, $\mathcal{G}^*$ is a forest (i.e., every of its vertices has at most one child), and $\mathcal{G}^*$ is a C-component then $\mathcal{G}^*$ is called a C-forest. The vertices which have no children are called roots and we say a C-forest is $\mathbb{R}^*$-rooted if it has roots $\mathbb{R}^* \subseteq \mathbb{V}^*$*

**Definition 10** (Hedge, Shpitser & Pearl (2006; 2008)). *Consider a graph $\mathcal{G}^* = (\mathbb{V}^*, \mathbb{E}^*)$ (whether it be a C-DMG or an ADMG) and two disjoint sets of vertices $\mathbb{X}^*, \mathbb{Y}^* \subseteq \mathbb{V}^*$. Let $\mathbb{F} = (\mathbb{V}^*_\mathbb{F}, \mathbb{E}^*_\mathbb{F})$ and $\mathbb{F}' = (\mathbb{V}^*_{\mathbb{F}'}, \mathbb{E}^*_{\mathbb{F}'})$ be*

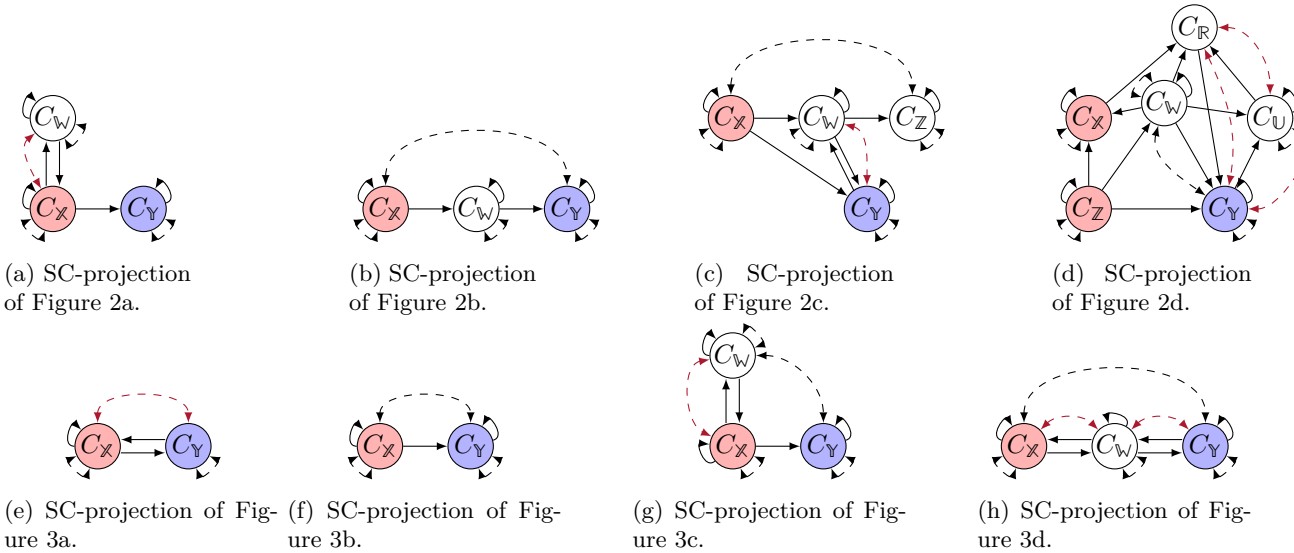

Figure 4: SC-projections of the C-DMGs in Figures 1, 2, and Figures 3. Each pair of red and blue vertices represents the total effect we are interested in, and the red edges indicate those added through the SC-projection.

two $\mathbb{R}^*$-rooted C-forests subgraphs of $\mathcal{G}^*$ such that $\mathbb{X}^* \cap \mathbb{V}_{\mathbb{F}}^* \neq \varnothing$, $\mathbb{X}^* \cap \mathbb{V}_{\mathbb{F}'}^* = \varnothing$, $\mathbb{F}' \subseteq \mathbb{F}$, and $\mathbb{R}^* \subset An(\mathbb{Y}^*, \mathcal{G}_{\overline{\mathbb{X}^*}}^*)$. Then $\mathbb{F}$ and $\mathbb{F}'$ form a hedge for the pair $(\mathbb{X}^*, \mathbb{Y}^*)$ in $\mathcal{G}^*$.

A hedge turned out to be too weak to cover non-identifiability in C-DMGs. For example, the C-DMG in Figure 3a contains no hedge but the macro causal effect is not identifiable due to the cycle between $\mathbb{C}_{\mathbb{X}}$ and $\mathbb{C}_{\mathbb{Y}}$. However, the SC-hedge—an extension of the hedge structure introduced in Ferreira & Assaad (2025) for summary causal graphs—proves to be applicable to general C-DMGs. In the following, we formally define SC-hedges in the context of C-DMGs and demonstrate that this substructure serves as a sound criterion for detecting non-identifiable macro causal effects.

**Definition 11** (Strongly connected projection (SC-projection)). *Consider a C-DMG $\mathcal{G}^{\mathbb{c}} = (\mathbb{C}, \mathbb{E}^{\mathbb{c}})$. The SC-projection $\mathcal{H}^{\mathbb{c}}$ of $\mathcal{G}^{\mathbb{c}}$ is the graph that includes all vertices and edges from $\mathcal{G}^{\mathbb{c}}$, plus a dashed bidirected edge between each pair $C_{\mathbb{X}}, C_{\mathbb{Y}} \in \mathbb{C}$ such that $Scc(C_{\mathbb{X}}, \mathcal{G}^{\mathbb{c}}) = Scc(C_{\mathbb{Y}}, \mathcal{G}^{\mathbb{c}})$ and $C_{\mathbb{X}} \neq C_{\mathbb{Y}}$.*

**Definition 12** (Strongly Connected Hedge (SC-Hedge)). *Consider a C-DMG $\mathcal{G}^{\mathbb{c}} = (\mathbb{C}, \mathbb{E}^{\mathbb{c}})$, its SC-projection $\mathcal{H}^{\mathbb{c}}$ and two disjoints sets of vertices $\mathbb{C}_{\mathbb{X}}, \mathbb{C}_{\mathbb{Y}} \subseteq \mathbb{C}$. A hedge for $(\mathbb{C}_{\mathbb{X}}, \mathbb{C}_{\mathbb{Y}})$ in $\mathcal{H}^{\mathbb{c}}$ is an SC-hedge for $(\mathbb{C}_{\mathbb{X}}, \mathbb{C}_{\mathbb{Y}})$ in $\mathcal{G}^s$.*

**Theorem 5.** *Consider a C-DMG $\mathcal{G}^{\mathbb{c}} = (\mathbb{C}, \mathbb{E}^{\mathbb{c}})$ and two disjoints sets of vertices $\mathbb{C}_{\mathbb{X}}, \mathbb{C}_{\mathbb{Y}} \subseteq \mathbb{C}$. Under Assumption 1, if there exists an SC-hedge for $(\mathbb{C}_{\mathbb{X}}, \mathbb{C}_{\mathbb{Y}})$ in $\mathcal{G}^{\mathbb{c}}$ then $\Pr(\mathbb{c}_{\mathbb{y}} \mid do(\mathbb{c}_{\mathbb{x}}))$ is not identifiable.*

Figure 4 illustrates all SC-projections of the C-DMGs presented in Figures 2 and 3. Notably, the SC-projections corresponding to the C-DMGs in Figure 2 do not contain a hedge, whereas all SC-projections derived from the C-DMGs in Figure 3 include a hedge. This distinction highlights the structural differences between the hedges and non-hedges and their implications for causal effect identification.

Note that Theorem 5 states the soundness of the SC-hedges or in other words it guarantees the non identifiability of the macro causal effect in the presence of a SC-hedge. However, it does not give the corresponding completeness result *i.e.*, the absence of an SC-hedge does not imply the identifiability of the macro causal effect.

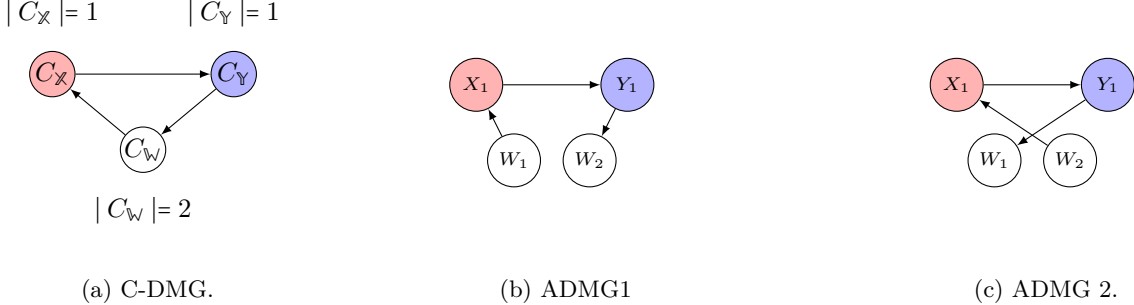

(a) C-DMG.  (b) ADMG1  (c) ADMG 2.

Figure 5: A C-DMG with the knowledge of the clusters' size and its only two compatible ADMGs that illustrates that knowing cluster sizes affects the completeness of d-separation and the do-calculus. The C-DMG in (a) contains two clusters of size 1 and it contains an SC-Hedge. However, even in the presence of an SC-Hedge the causal effect can be identifiable since in all its compatible ADMGs (*i.e.*, (b) and (c)) the causal effect remains identifiable by only applying Rule 2 of the do-calculus.

## 4 Non-completeness of d-separation and the do-calculus when considering the size of the clusters

Up to this point in the paper, we have assumed either that the size of each cluster is unknown or that no two adjacent clusters in a cycle are both of size one Assumption 1). In this section, we demonstrate that the completeness results of d-separation and the do-calculus no longer hold when cluster sizes are known and when some clusters contain only a single variable (when Assumption 1 is not satisfied). In the extreme cases where Assumption 1 is not verified, there exists some paths in the C-DMG which do not map to any micro-path in any compatible ADMG. To illustrate this, consider the C-DAG in Figure 5a, where cluster $C_{\mathbb{X}}$ has size 1, cluster $C_{\mathbb{Y}}$ also has size 1, and cluster $C_{\mathbb{W}}$ has size 2. In Figure 5a, the path $\langle C_{\mathbb{X}} \leftarrow C_{\mathbb{W}} \leftarrow C_{\mathbb{Y}} \rangle$ seems to be active, however when one looks at the two compatible ADMGs Figures 5b and 5c, one can see that there is actually no such path in the ADMGs. Using our SC-hedge characterization, we would conclude that the causal effect is not identifiable using the do-calculus, as the presence of a cycle containing $C_{\mathbb{X}}$ and $C_{\mathbb{Y}}$ implies the existence of a dashed bidirected edge between $C_{\mathbb{X}}$ and $C_{\mathbb{Y}}$ in the SC-Projection. However, in this specific case, the only compatible ADMGs are those shown in Figures 5b and 5c, where it is evident that the causal effect $\Pr(c_{\mathbb{y}} \mid \mathrm{do}(c_{\mathbb{x}})) = \Pr(y_1 \mid \mathrm{do}(x_1))$ is identifiable and equals $\Pr(y_1 \mid x_1)$. The key intuition behind this discrepancy is that a cycle involving $X_1$, $Y_1$, and $W_i$ (for $i \in \{1, 2\}$) cannot exist, as this would violate the assumption that the true causal structure follows an ADMG, where cycles are not permitted. While Assumption 1 is not required for the results concerning identifiability *i.e.*, Theorems 1 and 3, it is necessary for the results regarding non-identifiability to hold *i.e.*, Theorems 2, 4 and 5.

## 5 Conclusion and discussion

In this paper, we established the soundness and completeness of d-separation and the do-calculus for respectively identifying macro causal effects in C-DMGs. By doing so, we bridged the gap between causal inference and many real-world applications in epidemiology.

There are three main limitations to this work. The first limitation is that the completeness result in Theorem 4 does not take into account that there might exist different iterations of the rules of the do-calculus in different ADMGs that can give the same final identification of the causal effect. A second related limitation is that we provided a graphical characterization for the non-identifiability of macro causal effects, however this characterization is not proven to be complete, even though we did not find any counter-example of its completeness. Proving it complete remains an open problem. The third limitation is that our results rely on Assumption 1 which is not always satisfied in practice. However, we think that we can relax it, while keeping the same results, by assuming that no two adjacent nodes in a cycle are both of size 1.

**Acknowledgments**

We thank Fabrice Carrat from IPLESP for his valuable insights on the importance of C-DMGs in epidemiology. This work was supported by the CIPHOD project (ANR-23-CPJ1-0212-01).

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
