# A Appendix

## A.1 Useful definitions and properties

**Definition 13** (Primary Path (Ferreira & Assaad, 2024)). *Let $\mathcal{G}^* = (\mathbb{V}^*, \mathbb{E}^*)$ be a graph and $\tilde{\pi} = \langle V_1^*, \cdots, V_n^* \rangle$ a walk. The primary path of $\tilde{\pi}$ is noted $\tilde{\pi}_p$ and is defined iteratively as $U_1^* = V_1^*$, $\forall 1 \leq k, U_{k+1}^* = V_{max\{i|V_i^*=U_k^*\}+1}^*$ until $U_{k+1}^* = V_n^*$ with $\langle U_k^* \to U_{k+1}^* \rangle \subseteq \tilde{\pi}_p$ (resp. $\leftarrow$, $\leftarrow\dashrightarrow$) if $\langle V_{max\{i|V_i^*=U_k^*\}}^* \to V_{max\{i|V_i^*=U_k^*\}+1}^* \rangle \subseteq \tilde{\pi}$ (resp. $\leftarrow$, $\leftarrow\dashrightarrow$).*

**Property 1.** *Let $\mathcal{G}^* = (\mathbb{V}^*, \mathbb{E}^*)$ be a graph, $\tilde{\pi} = \langle V_1^*, \cdots, V_n^* \rangle$ a walk and $\tilde{\pi}_p = \langle U_1^*, \cdots, U_m^* \rangle$ its primary path.*

1. *$V_1^* = U_1^*$ and $V_n^* = U_m^*$, and*

2. *$\exists i : 1 \leq i < m$, $\langle U_i^* \to U_{i+1}^* \rangle \subseteq \tilde{\pi}_p$ (resp. $\leftarrow$, $\leftarrow\dashrightarrow$) $\implies$ $\exists j : 1 \leq j < n$, $U_i^* = V_j^*$, $U_{i+1}^* = V_{j+1}^*$, and $\langle V_j^* \to V_{j+1}^* \rangle \subseteq \tilde{\pi}$ (resp. $\leftarrow$, $\leftarrow\dashrightarrow$).*

*Proof.* This properties are directly given by Definition 13. $\square$

**Property 2** (Equivalence between blocking walks and paths). *Let $\mathcal{G}^* = (\mathbb{V}^*, \mathbb{E}^*)$ be a graph (whether it be an ADMG or a C-DMG), $\mathbb{W}^* \subseteq \mathbb{V}^*$ and $\tilde{\pi} = \langle V_1^*, \cdots, V_n^* \rangle$ be a walk. If $\tilde{\pi}$ is $\mathbb{W}^*$-active then its primary path $\tilde{\pi}_p$ is $\mathbb{W}^*$-active.*

*Proof.* Let $\mathcal{G}^* = (\mathbb{V}^*, \mathbb{E}^*)$ be a graph (whether it be an ADMG or a C-DMG), $\mathbb{W}^* \subseteq \mathbb{V}^*$ and $\tilde{\pi} = \langle V_1^*, \cdots, V_n^* \rangle$ be an $\mathbb{W}^*$-active walk. Suppose that its primary path $\tilde{\pi}_p = \langle U_1^*, \cdots, U_m^* \rangle$ is $\mathbb{W}^*$-blocked.

- If $U_1^* \in \mathbb{W}^*$ or $U_m^* \in \mathbb{W}^*$ then using Property 1 item 1, $V_1^* \in \mathbb{W}^*$ or $V_n^* \in \mathbb{W}^*$ and thus $\tilde{\pi}$ is blocked by $\mathbb{W}^*$ which contradicts the initial assumption.

Otherwise, take $1 < i < m$ such that $\langle U_{i-1}^*, U_i^*, U_{i+1}^* \rangle$ is $\mathbb{W}^*$-blocked.

- If $U_{i-1}^* \leftrightarrow U_i^* \to U_{i+1}^*$ (resp. $U_{i-1}^* \leftarrow U_i^* \leftrightarrow U_{i+1}^*$) and $U_i^* \in \mathbb{W}^*$ then, using Property 1 item 2, there exists $1 < j < n$ such that $U_i^* = V_j^*$ and $U_{i+1}^* = V_{j+1}^*$ (resp. $U_{i-1}^* = V_{j-1}^*$ and $U_i^* = V_j^*$) and thus $V_{j-1}^* \leftrightarrow V_j^* \to V_{j+1}^*$ (resp. $V_{j-1}^* \leftarrow V_j^* \leftrightarrow U_{j+1}^*$) is in $\tilde{\pi}$ and $V_j^* = U_i^* \in \mathbb{W}^*$. Therefore, $\tilde{\pi}$ is $\mathbb{W}^*$-blocked which contradicts the initial assumption.

- If $U_{i-1}^* \leftrightarrow U_i^* \leftrightarrow U_{i+1}^*$ and $De(U_i^*, \mathcal{G}^*) \cap \mathbb{W}^* = \varnothing$ then, using Property 1 item 2, there exists $1 < j < n$ such that $U_{i-1}^* = V_{j-1}^*$ and $U_i^* = V_j^*$ and $V_{j-1}^* \leftrightarrow V_j^* \cdots \leftrightarrow$ is in $\tilde{\pi}$. Take $k_{min} = min\{j \leq k < n \mid \langle V_{k-1}^* \leftrightarrow V_k^* \leftrightarrow V_{k+1}^* \rangle \subseteq \tilde{\pi}\}$ and notice that $De(V_{k_{min}}^*, \mathcal{G}^*) \subseteq De(V_j^*, \mathcal{G}^*) = De(U_i^*, \mathcal{G}^*)$ so $De(V_{k_{min}}^*, \mathcal{G}^*) \cap \mathbb{W}^* = \varnothing$. Therefore, $\tilde{\pi}$ is $\mathbb{W}^*$-blocked which contradicts the initial assumption.

In conclusion, for any $\mathbb{W}^* \subseteq \mathbb{V}^*$, if a walk is $\mathbb{W}^*$-active then its primary path is $\mathbb{W}^*$-active as well. $\square$

In the following, we introduce a property of compatibility between mutilated graphs that will be useful for proving the soundness and the completeness of the do-calculus in C-DMGs.

**Property 3.** *(Compatibility of Mutilated Graphs) Let $\mathcal{G} = (\mathbb{V}, \mathbb{E})$ be an ADMG, $\mathcal{G}^c = (\mathbb{C}, \mathbb{E}^s)$ its compatible C-DMG and $\mathbb{C}_\mathbb{A}, \mathbb{C}_\mathbb{B} \subseteq \mathbb{C}$. The mutilated graph $\mathcal{G}^c_{\overline{\mathbb{C}_\mathbb{A}}\underline{\mathbb{C}_\mathbb{B}}}$ is a C-DMG compatible with the mutilated ADMG $\mathcal{G}_{\overline{\mathbb{A}}\underline{\mathbb{B}}}$ where $\mathbb{A} = \bigcup_{C \in \mathbb{C}_\mathbb{A}} C$ and $\mathbb{B} = \bigcup_{C \in \mathbb{C}_\mathbb{B}} C$.*

*Proof.* Let $\mathcal{G} = (\mathbb{V}, \mathbb{E})$ be an ADMG, $\mathcal{G}^c = (\mathbb{C}, \mathbb{E}^c)$ its compatible C-DMG, $\mathbb{C}_\mathbb{A}, \mathbb{C}_\mathbb{B} \subseteq \mathbb{C}$, $\mathbb{A} = \bigcup_{C \in \mathbb{C}_\mathbb{A}} C$ and $\mathbb{B} = \bigcup_{C \in \mathbb{C}_\mathbb{B}} C$.

Firstly, let us show that every arrow in $\mathcal{G}^c_{\overline{\mathbb{C}_\mathbb{A}}\underline{\mathbb{C}_\mathbb{B}}}$ corresponds to an arrow in $\mathcal{G}_{\overline{\mathbb{A}}\underline{\mathbb{B}}}$. Let $C, C' \in \mathbb{C}$ such that $C \to C'$ is in $\mathbb{E}^c_{\overline{\mathbb{A}}\underline{\mathbb{B}}}$. We know that $C \notin \mathbb{C}_\mathbb{B}$ and $C' \notin \mathbb{C}_\mathbb{A}$ and that $C \to C'$ is in $\mathbb{E}^c$. Therefore, there exists $V \in C, V' \in C'$ such that $V \to V'$ is in $\mathbb{E}$ and $V \notin \mathbb{B}$ and $V' \notin \mathbb{A}$ so $V \to V'$ is in $\mathbb{E}_{\overline{\mathbb{A}}\underline{\mathbb{B}}}$.

Similarly, let $C, C' \in \mathbb{C}$ such that $C \leftrightarrow C'$ is in $\mathbb{E}^{\mathfrak{c}}_{\overline{\mathbb{C}_{\mathbb{A}}}\underline{\mathbb{C}_{\mathbb{B}}}}$. We know that $C, C' \notin \mathbb{C}_{\mathbb{A}}$ and that $C \leftrightarrow C'$ is in $\mathbb{E}^{\mathfrak{c}}$. Therefore, there exists $V \in C, V' \in C'$ such that $V \leftrightarrow V'$ is in $\mathbb{E}$ and $V, V' \notin \mathbb{A}$ so $V \leftrightarrow V'$ is in $\mathbb{E}_{\overline{\mathbb{A}}\underline{\mathbb{B}}}$.

Secondly, let us show that every arrow in $\mathcal{G}_{\overline{\mathbb{A}}\underline{\mathbb{B}}}$ corresponds to an arrow in $\mathcal{G}^{\mathfrak{c}}_{\overline{\mathbb{C}_{\mathbb{A}}}\underline{\mathbb{C}_{\mathbb{B}}}}$. Let $V, V' \in \mathbb{V}_{\overline{\mathbb{A}}\underline{\mathbb{B}}}$ such that $V \to V'$ is in $\mathbb{E}_{\overline{\mathbb{A}}\underline{\mathbb{B}}}$. Let $C = Cl(V, \mathcal{G}^{\mathfrak{c}})$ and $C' = Cl(V', \mathcal{G}^{\mathfrak{c}})$. We know that $V \notin \mathbb{B}$ and $V' \notin \mathbb{A}$ and that $V \to V'$ is in $\mathbb{E}$. Therefore, $C \to C'$ is in $\mathbb{E}^{\mathfrak{c}}$ and $C \notin \mathbb{C}_{\mathbb{B}}$ and $V' \notin \mathbb{C}_{\mathbb{A}}$ so $C \to C'$ is in $\mathbb{E}^{\mathfrak{c}}_{\overline{\mathbb{C}_{\mathbb{A}}}\underline{\mathbb{C}_{\mathbb{B}}}}$.

Similarly, let $V, V' \in \mathbb{V}_{\overline{\mathbb{A}}\underline{\mathbb{B}}}$ such that $V \leftrightarrow V'$ is in $\mathbb{E}_{\overline{\mathbb{A}}\underline{\mathbb{B}}}$. Let $C = Cl(V, \mathcal{G}^{\mathfrak{c}})$ and $C' = Cl(V', \mathcal{G}^{\mathfrak{c}})$ We know that $V, V' \notin \mathbb{A}$ and that $V \leftrightarrow V'$ is in $\mathbb{E}$. Therefore, $C \leftrightarrow C'$ is in $\mathbb{E}^{\mathfrak{c}}$ and $C, C' \notin \mathbb{C}_{\mathbb{A}}$ so $V \leftrightarrow V'$ is in $\mathbb{E}^{\mathfrak{c}}_{\overline{\mathbb{C}_{\mathbb{A}}}\underline{\mathbb{C}_{\mathbb{B}}}}$. □

## A.2 Proof of Theorem 1

*Proof.* Suppose $\mathbb{C}_{\mathbb{X}}$ and $\mathbb{C}_{\mathbb{Y}}$ are d-separated by $\mathbb{C}_{\mathbb{W}}$ in $\mathcal{G}^{\mathfrak{c}}$ and there exists a compatible ADMG $\mathcal{G} = (\mathbb{V}, \mathbb{E})$ and a path $\pi = \langle V_1, \cdots, V_n \rangle$ in $\mathcal{G}$ from $V_1 \in \mathbb{X}$ to $V_n \in \mathbb{Y}$ which is not blocked by $\mathbb{W}$. Consider the walk $\tilde{\pi} = \langle C_1, \cdots, C_n \rangle$ with $\forall 1 \le i \le n, C_i = Cl(V_i, \mathcal{G}^{\mathfrak{c}})$ and $\forall 1 \le i < n, \langle C_i \to C_{i+1} \rangle \subseteq \tilde{\pi}$ (resp. $\leftarrow, \leftrightarrow$) $\iff \langle V_i \to V_{i+1} \rangle \subseteq \pi$ (resp. $\leftarrow, \leftrightarrow$). $\tilde{\pi}$ is a walk from $\mathbb{C}_{\mathbb{X}}$ to $\mathbb{C}_{\mathbb{Y}}$ in $\mathcal{G}^{\mathfrak{c}}$. Since $\mathbb{C}_{\mathbb{X}}$ and $\mathbb{C}_{\mathbb{Y}}$ are d-separated by $\mathbb{C}_{\mathbb{W}}$, using the contraposition of Property 2, we know that $\mathbb{C}_{\mathbb{W}}$ blocks $\tilde{\pi}$.

- If $C_1 \in \mathbb{W}$ or $C_n \in \mathbb{C}_{\mathbb{W}}$, then $V_1 \in \mathbb{W}$ or $V_n \in \mathbb{W}$ and thus $\pi$ is blocked by $\mathbb{W}$ which contradicts the initial assumption.

Otherwise, take $1 < i < n$ such that $\langle C_{i-1}, C_i, C_{i+1} \rangle$ is $\mathbb{C}_{\mathbb{W}}$-blocked.

- If $\langle C_{i-1} \leftrightarrow C_i \to C_{i+1} \rangle \subseteq \tilde{\pi}$ (resp. $\langle C_{i-1} \leftarrow C_i \leftrightarrow C_{i+1} \rangle \subseteq \tilde{\pi}$) and $C_i \in \mathbb{C}_{\mathbb{W}}$ then, $\langle V_{i-1} \leftrightarrow V_i \to V_{i+1} \rangle \subseteq \pi$ (resp. $\langle V_{i-1} \leftarrow V_i \leftrightarrow V_{i+1} \rangle \subseteq \pi$) and $V_i \in C_i \subseteq \mathbb{W} = \bigcup_{C \in \mathbb{C}_{\mathbb{W}}}$. Thus $\pi$ is blocked by $\mathbb{W}$ which contradicts the initial assumption.

- If $\langle C_{i-1} \leftrightarrow C_i \leftrightarrow C_{i+1} \rangle \subseteq \tilde{\pi}$ and $De(C_i, \mathcal{G}^{\mathfrak{c}}) \cap \mathbb{C}_{\mathbb{W}} = \varnothing$ then, $\langle V_{i-1} \leftrightarrow V_i \leftrightarrow V_{i+1} \rangle \subseteq \pi$. Moreover, $De(V_i, \mathcal{G}) \subseteq \bigcup_{C \in De(C_i, \mathcal{G}^{\mathfrak{c}})} C$ so $De(V_i, \mathcal{G}) \cap \mathbb{W} \subseteq \bigcup_{C \in De(C_i, \mathcal{G}^{\mathfrak{c}}) \cap \mathbb{C}_{\mathbb{W}}} C$ and $De(C_i, \mathcal{G}^{\mathfrak{c}}) \cap \mathbb{C}_{\mathbb{W}} = \varnothing$ thus $De(V_i, \mathcal{G}) \cap \mathbb{W} = \varnothing$. Therefore, $\pi$ is blocked by $\mathbb{W}$ which contradicts the initial assumption.

In conclusion, d-separation is sound in C-DMGs.

□

## A.3 Proof of Theorem 2

*Proof.* Suppose $\mathbb{C}_{\mathbb{X}}$ and $\mathbb{C}_{\mathbb{Y}}$ are not d-separated by $\mathbb{C}_{\mathbb{W}}$ in $\mathcal{G}^{\mathfrak{c}}$. There exists an $\mathbb{C}_{\mathbb{W}}$-active path $\pi = \langle C_1, \cdots, C_n \rangle$ with $C_1 \in \mathbb{C}_{\mathbb{X}}$ and $C_n \in \mathbb{C}_{\mathbb{Y}}$. Because of Assumption 1 we either have no information regarding the size of the clusters, and thus we can assume that every cluster is of size at least 2, either no two adjacent cluster in a cycle are both of size 1. Let $\le_D$ be a total order on $\mathbb{C}$ such that for every collider $C_i$ in $\pi$ there exists a descendant path from $C_i$ to $\mathbb{C}_{\mathbb{W}}$ which is compatible with the order $\le_D$ (*i.e.*, $\forall 1 < i < n, \langle C_{i-1} \leftrightarrow C_i \leftrightarrow C_{i+1} \rangle \subseteq \pi, \exists \pi_D = \langle D_1 \to \cdots \to D_m \rangle$ such that $D_1 = C_i, D_m \in \mathbb{C}_{\mathbb{W}}$ and $\forall 1 \le j < m, D_j \le_D D_{j+1}$). From the C-DMG $\mathcal{G}^{\mathfrak{c}} = (\mathbb{C}, \mathbb{E}^{\mathfrak{c}})$ one can build a compatible ADMG $\mathcal{G} = (\mathbb{V}, \mathbb{E})$ in the following way:

- For every cluster $C \in \mathbb{C}$ consider two variable $V_C^0, V_C^1 \in C$ with $V_C^0 = V_C^1$ if and only if $|C| = 1$.

$$\mathbb{E}^{\to} := \{V_C^0 \to V_{C'}^1 \mid \forall C \to C' \in \mathbb{E}^{\mathfrak{c}}\}$$
$$\mathbb{E}_{\pi}^{\to} := \{V_C^0 \to V_{C'}^0 \mid \forall C \to C' \in \mathbb{E}^{\mathfrak{c}}$$
$$\text{such that } \langle C \to C' \rangle \subseteq \pi \text{ or } \langle C' \leftarrow C \rangle \subseteq \pi\}$$
$$\mathbb{E}_D^{\to} := \{V_C^1 \to V_{C'}^1 \mid \forall C \to C' \in \mathbb{E}^{\mathfrak{c}}$$
$$\text{such that } C \le_D C'\}$$
$$\mathbb{E}^{\leftrightarrow} := \{V_C^0 \to V_{C'}^0 \mid \forall C \leftrightarrow C' \in \mathbb{E}^{\mathfrak{c}}\}$$
$$\mathbb{E} := \mathbb{E}^{\to} \cup \mathbb{E}_{\pi}^{\to} \cup \mathbb{E}_D^{\to} \cup \mathbb{E}^{\leftrightarrow}$$

Notice that $\mathcal{G}$ is in indeed acyclic and is compatible with the C-DMG $\mathcal{G}^{\mathbb{c}}$. Moreover, $\mathcal{G}$ contains the path $\pi_{\mathcal{G}} = \langle V_{C_1}^0, \cdots, V_{C_n}^0 \rangle$ which is necessarily $\mathbb{W}$-active since $\pi$ is $\mathbb{C}_{\mathbb{W}}$-active. In conclusion, d-separation in C-DMGs is complete under Assumption 1. $\qquad\square$

## A.4 Proof of Theorem 3

*Proof.* Let $\mathcal{G}^{\mathbb{c}} = (\mathbb{C}, \mathbb{E}^{\mathbb{c}})$ be a C-DMG and $\mathbb{C}_{\mathbb{X}}, \mathbb{C}_{\mathbb{Y}}, \mathbb{C}_{\mathbb{Z}}, \mathbb{C}_{\mathbb{W}} \subseteq \mathbb{C}$ be disjoints subsets of vertices. Let $\mathbb{X} = \bigcup_{C \in \mathbb{C}_{\mathbb{X}}} C$, $\mathbb{Y} = \bigcup_{C \in \mathbb{C}_{\mathbb{Y}}} C$, $\mathbb{Z} = \bigcup_{C \in \mathbb{C}_{\mathbb{Z}}} C$ and $\mathbb{W} = \bigcup_{C \in \mathbb{C}_{\mathbb{W}}} C$.

Suppose that $(\mathbb{C}_{\mathbb{Y}} \perp\!\!\!\perp_d \mathbb{C}_{\mathbb{X}} \mid \mathbb{C}_{\mathbb{Z}}, \mathbb{C}_{\mathbb{W}})_{\mathcal{G}^{\mathbb{c}}_{\overline{\mathbb{C}_{\mathbb{Z}}}}}$. Using Property 3 and Theorem 1 we know that for every compatible ADMG $\mathcal{G}$, $(\mathbb{Y} \perp\!\!\!\perp_d \mathbb{X} \mid \mathbb{Z}, \mathbb{W})_{\mathcal{G}_{\overline{\mathbb{Z}}}}$. Thus, the usual rule 1 of the do-calculus given by Pearl 1995 in the case of ADMGs states that $\Pr(\mathbb{y} \mid \text{do}(\mathbb{z}), \mathbb{x}, \mathbb{w}) = \Pr(\mathbb{y} \mid \text{do}(\mathbb{z}), \mathbb{w})$.

Suppose that $(\mathbb{C}_{\mathbb{Y}} \perp\!\!\!\perp_d \mathbb{C}_{\mathbb{X}} \mid \mathbb{C}_{\mathbb{Z}}, \mathbb{C}_{\mathbb{W}})_{\mathcal{G}^{\mathbb{c}}_{\overline{\mathbb{C}_{\mathbb{Z}}}\underline{\mathbb{C}_{\mathbb{X}}}}}$. Using Property 3 and Theorem 1 we know that for every compatible ADMG $\mathcal{G}$, $(\mathbb{Y} \perp\!\!\!\perp_d \mathbb{X} \mid \mathbb{Z}, \mathbb{W})_{\mathcal{G}_{\overline{\mathbb{Z}}\underline{\mathbb{X}}}}$. Thus, the usual rule 2 of the do-calculus given by Pearl 1995 in the case of ADMGs states that $\Pr(\mathbb{y} \mid \text{do}(\mathbb{z}), \text{do}(\mathbb{x}), \mathbb{w}) = \Pr(\mathbb{y} \mid \text{do}(\mathbb{z}), \mathbb{x}, \mathbb{w})$.

Suppose that $(\mathbb{C}_{\mathbb{Y}} \perp\!\!\!\perp_d \mathbb{C}_{\mathbb{X}} \mid \mathbb{C}_{\mathbb{Z}}, \mathbb{C}_{\mathbb{W}})_{\mathcal{G}^{\mathbb{c}}_{\overline{\mathbb{C}_{\mathbb{Z}}\mathbb{C}_{\mathbb{X}}(\mathbb{C}_{\mathbb{W}})}}}$. Using Property 3, $\mathcal{G}^{\mathbb{c}}_{\overline{\mathbb{C}_{\mathbb{Z}}}}$ is compatible with $\mathcal{G}_{\overline{\mathbb{Z}}}$ and if $\exists X \in \mathbb{X}$ such that $X \in An(\mathbb{W}, \mathcal{G}_{\overline{\mathbb{Z}}})$ then $Cl(X, \mathcal{G}^{\mathbb{c}}) \in An(\mathbb{C}_{\mathbb{W}}, \mathcal{G}^{\mathbb{c}}_{\overline{\mathbb{C}_{\mathbb{Z}}}})$. Therefore, $\forall C_X \in \mathbb{C}_{\mathbb{X}},\ C_X \notin An(\mathbb{C}_{\mathbb{W}}, \mathcal{G}^{\mathbb{c}}_{\overline{\mathbb{C}_{\mathbb{Z}}}}) \implies \forall X \in \mathbb{X},\ X \notin An(\mathbb{X}, \mathcal{G}_{\overline{\mathbb{Z}}})$. Thus, $\mathcal{G}^{\mathbb{c}}_{\overline{\mathbb{C}_{\mathbb{Z}}\mathbb{C}_{\mathbb{X}}(\mathbb{C}_{\mathbb{W}})}}$ and $\mathcal{G}_{\overline{\mathbb{Z}\mathbb{X}(\mathbb{W})}}$ are compatible, and using Theorem 1 we know that for every compatible ADMG $\mathcal{G}$, $(\mathbb{Y} \perp\!\!\!\perp_d \mathbb{X} \mid \mathbb{Z}, \mathbb{W})_{\mathcal{G}_{\overline{\mathbb{Z}\mathbb{X}(\mathbb{W})}}}$. Thus, the usual rule 3 of the do-calculus given by Pearl 1995 in the case of ADMGs states that $\Pr(\mathbb{y} \mid \text{do}(\mathbb{z}), \text{do}(\mathbb{x}), \mathbb{w}) = \Pr(\mathbb{y} \mid \text{do}(\mathbb{z}), \mathbb{w})$. $\qquad\square$

## A.5 Proof of Theorem 4

*Proof.* Let $\mathcal{G}^{\mathbb{c}} = (\mathbb{C}, \mathbb{E}^{\mathbb{c}})$ be a C-DMG and $\mathbb{C}_{\mathbb{X}}, \mathbb{C}_{\mathbb{Y}}, \mathbb{C}_{\mathbb{Z}}, \mathbb{C}_{\mathbb{W}} \subseteq \mathbb{C}$.

Suppose that rule 1 does not apply *i.e.*, $(\mathbb{C}_{\mathbb{Y}} \not\perp\!\!\!\perp_d \mathbb{C}_{\mathbb{X}} \mid \mathbb{C}_{\mathbb{Z}}, \mathbb{C}_{\mathbb{W}})_{\mathcal{G}^{\mathbb{c}}_{\overline{\mathbb{C}_{\mathbb{Z}}}}}$. Then, using Theorem 2 there exists an ADMG $\tilde{\mathcal{G}}$ compatible with $\mathcal{G}^{\mathbb{c}}_{\overline{\mathbb{C}_{\mathbb{Z}}}}$ in which $(\mathbb{Y} \not\perp\!\!\!\perp_d \mathbb{X} \mid \mathbb{Z}, \mathbb{W})_{\tilde{\mathcal{G}}}$. Notice that there exists an ADMG $\mathcal{G}$ compatible with $\mathcal{G}^{\mathbb{c}}$ such that $\mathcal{G}_{\overline{\mathbb{Z}}} = \tilde{\mathcal{G}}$. Therefore, $\mathcal{G}$ is an ADMG compatible with $\mathcal{G}^{\mathbb{c}}$ in which $(\mathbb{Y} \not\perp\!\!\!\perp_d \mathbb{X} \mid \mathbb{Z}, \mathbb{W})_{\mathcal{G}_{\overline{\mathbb{Z}}}}$ and thus the usual rule 1 of the do-calculus given by Pearl 1995 in the case of ADMGs does not apply.

Suppose that rule 2 does not apply *i.e.*, $(\mathbb{C}_{\mathbb{Y}} \not\perp\!\!\!\perp_d \mathbb{C}_{\mathbb{X}} \mid \mathbb{C}_{\mathbb{Z}}, \mathbb{C}_{\mathbb{W}})_{\mathcal{G}^{\mathbb{c}}_{\overline{\mathbb{C}_{\mathbb{Z}}}\underline{\mathbb{C}_{\mathbb{X}}}}}$. Then, using Theorem 2 there exists an ADMG $\tilde{\mathcal{G}}$ compatible with $\mathcal{G}^{\mathbb{c}}_{\overline{\mathbb{C}_{\mathbb{Z}}}\underline{\mathbb{C}_{\mathbb{X}}}}$ in which $(\mathbb{Y} \not\perp\!\!\!\perp_d \mathbb{X} \mid \mathbb{Z}, \mathbb{W})_{\tilde{\mathcal{G}}}$. Notice that there exists an ADMG $\mathcal{G}$ compatible with $\mathcal{G}^{\mathbb{c}}$ such that $\mathcal{G}_{\overline{\mathbb{Z}}\underline{\mathbb{X}}} = \tilde{\mathcal{G}}$. Therefore, $\mathcal{G}$ is an ADMG compatible with $\mathcal{G}^{\mathbb{c}}$ in which $(\mathbb{Y} \not\perp\!\!\!\perp_d \mathbb{X} \mid \mathbb{Z}, \mathbb{W})_{\mathcal{G}_{\overline{\mathbb{Z}}\underline{\mathbb{X}}}}$ and thus the usual rule 2 of the do-calculus given by Pearl 1995 in the case of ADMGs does not apply.

Suppose that rule 3 does not apply *i.e.*, $(\mathbb{C}_{\mathbb{Y}} \not\perp\!\!\!\perp_d \mathbb{C}_{\mathbb{X}} \mid \mathbb{C}_{\mathbb{Z}}, \mathbb{C}_{\mathbb{W}})_{\mathcal{G}^{\mathbb{c}}_{\overline{\mathbb{C}_{\mathbb{Z}}\mathbb{C}_{\mathbb{X}}(\mathbb{C}_{\mathbb{W}})}}}$. Then, using Theorem 2 there exists an ADMG $\tilde{\mathcal{G}}$ compatible with $\mathcal{G}^{\mathbb{c}}_{\overline{\mathbb{C}_{\mathbb{Z}}\mathbb{C}_{\mathbb{X}}(\mathbb{C}_{\mathbb{W}})}}$ in which $(\mathbb{Y} \not\perp\!\!\!\perp_d \mathbb{X} \mid \mathbb{Z}, \mathbb{W})_{\tilde{\mathcal{G}}}$. Using the same idea as in the proof of Theorem 3, notice that there exists an ADMG $\mathcal{G}$ compatible with $\mathcal{G}^{\mathbb{c}}$ such that $\mathcal{G}_{\overline{\mathbb{Z}\mathbb{X}(\mathbb{W})}} = \tilde{\mathcal{G}}$. Therefore, $\mathcal{G}$ is an ADMG compatible with $\mathcal{G}^{\mathbb{c}}$ in which $(\mathbb{Y} \not\perp\!\!\!\perp_d \mathbb{X} \mid \mathbb{Z}, \mathbb{W})_{\mathcal{G}_{\overline{\mathbb{Z}\mathbb{X}(\mathbb{W})}}}$ and thus the usual rule 3 of the do-calculus given by Pearl 1995 in the case of ADMGs does not apply. $\qquad\square$

## A.6 Proof of Theorem 5

*Proof.* Consider an C-DMG $\mathcal{G}^{\mathbb{c}} = (\mathbb{C}, \mathbb{E}^{\mathbb{c}})$, its SC-projection $\mathcal{H}^{\mathbb{c}}$ and a SC-Hedge $\mathbb{F}, \mathbb{F}'$ for $\Pr(\mathbb{c}_{\mathbb{y}} \mid \text{do}(\mathbb{c}_{\mathbb{x}}))$. Let us prove Theorem 5 by induction on the number of bidirected dashed edges in the C-forest $\mathbb{F}$ that are in $\mathcal{H}^{\mathbb{c}}$ but not in $\mathcal{G}^{\mathbb{c}}$ (*i.e.*, which are artificially induced by cycles). $\forall C \in \mathbb{C}$, take $V_C \in C$.

Firstly, if $\mathbb{F}, \mathbb{F}'$ is a Hedge in $\mathcal{G}^{\mathbb{c}}$ then there exists a compatible ADMG $\mathcal{G} = (\mathbb{V}, \mathbb{E})$ such that $C \to C' \in \mathbb{E}^{\mathbb{F}'}$ (resp. $\leftarrow\!\!\dashrightarrow$) $\implies V_C \to V_{C'} \in \mathbb{E}$ (resp. $\leftarrow\!\!\dashrightarrow$) and thus if $\mathbb{F}_{\mathcal{G}} = (\{V_C \mid C \in \mathbb{V}^{\mathbb{F}}\}, \{V_C \to V_{C'} \mid C \to C' \in \mathbb{E}^{\mathbb{F}}\} \cup \{V_C \leftarrow\!\!\dashrightarrow V_{C'} \mid C \leftarrow\!\!\dashrightarrow C' \in \mathbb{E}^{\mathbb{F}}\})$ and $\mathbb{F}'_{\mathcal{G}} = (\{V_C \mid C \in \mathbb{V}^{\mathcal{F}'}\}, \{V_C \to V_{C'} \mid C \to C' \in \mathbb{E}^{\mathcal{F}'}\} \cup \{V_C \leftarrow\!\!\dashrightarrow V_{C'} \mid C \leftarrow\!\!\dashrightarrow C' \in \mathbb{E}^{\mathcal{F}'}\})$ is a Hedge for $\Pr(\mathbb{y} \mid \text{do}(\mathbb{x}))$ and thus $\Pr(\mathbb{c}_{\mathbb{y}} \mid \text{do}(\mathbb{c}_{\mathbb{x}})) = \Pr(\mathbb{y} \mid \text{do}(\mathbb{x}))$ is not identifiable.

Secondly, if there exists $C_X \in \mathbb{C}_{\mathbb{X}}$ and $C_Y \in \mathbb{C}_{\mathbb{Y}}$ such that $Scc(C_X, \mathcal{G}^{\mathfrak{c}}) = Scc(C_Y, \mathcal{G}^{\mathfrak{c}})$ then, because of Assumption 1, there exists a compatible ADMG in which there is a directed path from $V_{C_Y}$ to $V_{C_X}$. This path must be blocked as it is not causal but every vertex on this path is a descendant of $V_{C_Y}$ and thus cannot be adjusted on without inducing a bias. Lastly, assume that Theorem 5 is true for any SC-Hedge with $k$ bidirected dashed edges which are in $\mathcal{H}^{\mathfrak{c}}$ but not in $\mathcal{G}^{\mathfrak{c}}$ and that $\mathbb{F}$ has $k+1$ such edges. Then, one cannot identify the macro causal effect by the adjustment formula Shpitser et al. (2010) due to the ambiguity (Assaad et al., 2024) induced by the cycle on $C_X$ or due to the bias induced by the latent confounder of $C_X$ that cannot be removed without using any rule of the do-calculus. Moreover, any decomposition of the effect using the do-calculus will necessitate the identification of other macro causal effects. These effects have sub-C-forests of $\mathbb{F}, \mathbb{F}'$ as SC-Hedges and at least one of theses sub-C-forests has at most $k$ artificial bidirected dashed edges induced by cycles. The associated macro causal effect is therefore unidentifiable by induction. □