# OpenReview forum: "Identifying Macro Causal Effects in a C-DMG over ADMGs"
_TMLR — Accepted by TMLR_

### Review · Reviewer_QfrF · 2025-05-16

**Summary Of Contributions:**

This paper investigates the soundness and completeness of d-separation and do-calculus in cluster directed mixed graphs, with a focus on macro-level effects.

**Audience:**

Yes

**Broader Impact Concerns:**

I do not think this paper needs to address broader impact concerns.

**Claims And Evidence:**

Yes

**Requested Changes:**

**Minor comments**

Page 3, 1st paragraph, 5th line: "two function" -> "two functions"

Page 4, 1st paragraph, 2nd line: "implies in all" -> "implies that in all"

Page 4, 2nd paragraph, 4th line: "a ADMG" -> "an ADMG". Also in Definition 6 and Definition 7.

Page 5, Definition 5: Are $\mathbb{X}$ and $\mathbb{Y}$ supposed to represent clusters in $\mathcal{G}^{\mathbb{C}}$? I think this should be clearly stated either way. Also, what do you mean by a positive distribution? Do you mean that the densities are positive everywhere? Then you should have mentioned from the beginning that you are assuming existence of densities.

Page 6, Theorem 1, 2nd line: "separated" -> "d-separated"

Page 6, Example 1, 2nd line: "in in" -> "in"

Page 6, Footnote 2, 2nd line: "cluster were" -> "clusters were". On the 4th line of this footnote, the first "nor" should be "neither".

Page 6, Section 3.2, 2nd line: "i.e., that contains" -> "i.e., those that contain"

Page 7, Theorem 3, last line: the subscript should contain a bar?

Page 7 in general: "the do-calculus" -> "do-calculus"

Page 8, paragraph below Example 5: Again, you never said what the positivity assumption is, nor did you give a reference for this.

Page 8, section 3.3, 1st line: "an hedge" -> "a hedge". Also in Definition 10, just below Definition 10, Definition 12

Page 8, Definition 10: The authors are using $\mathbb{F}$ and $\mathcal{F}$ interchangeably here. Please be consistent.

Page 8, Definition 10: In (Shpitser and Pearl), the definition of Hedges are such that $\mathbb{R}^*\subset An(\mathbb{Y}^*,\mathcal{G}^*_{\bar{\mathbb{X}^*}})$, but here the bar is under the $\mathbb{X}^*$. Is this a typo?

Page 9, Definition 12 & Theorem 5, 1st line: "an C-DMG" -> "a C-DMG"

**Strengths And Weaknesses:**

** Strengths **

This is a solid paper where I can clearly see the motivation, and the mathematical developments appear to be sound. I am familiar with causality in general but I was not familiar with this particularly line of work, but I am convinced that this paper proves natural and important extensions of famous results in identifiability to cluster graphs. Identifiability is one of the most important problems in the field of causal inference (arguably it is the domain-defining problem). Therefore, I recommend to accept this paper.

** Weaknesses **

In the authors' opinion, how strong / restrictive is the assumption that each exogenous variable causally affects at most two endogenous variables? It seems quite strong to me.

Also, while this is a solid paper, the novelty is not huge, as the results appear to be straightforward extensions of well-known classical results in DAGs, especially those in (Shpitser and Pearl, 2008). However, the TMLR evaluation criteria explicitly recommends to accept papers of this kind, so I will recommend to accept.

---

> ### Author Response · Authors · 2025-05-23
>
> We appreciate the reviewer for their thorough review of our paper and for offering valuable comments and suggestions.
>
> How strong/weak is Assumption 1?
> -Indeed, the assumption (as written) is slightly too strong. However, we wrote it in such a way for simplicity but one could replace it with "Every cluster which is in a cycle is of size at least 2" or even "No two adjacent clusters in a cycle in the C-DMG are both of size 1" or more formally $\forall \langle C_1\rightarrow\cdots\rightarrow C_n\rangle \in \mathcal{G}^\mathbb{c}$, with $C_1=C_n:\forall 1\leq i <n,|C_i|>1$ or $|C_{i+1}|>1$.
> As detailed to Reviewer p9DU, the purpose of Assumption 1 is to guarantee that if a directed path $\langle C_1 \rightarrow \cdots \rightarrow C_n\rangle$ exists in the C-DMG then there exists an ADMG in which the directed path $\langle V_1 \rightarrow \cdots \rightarrow V_n \rangle$ exists with $\forall i,~V_i\in C_i$.
> Therefore, we do believe that the actual assumption needed is fairly weak.
> We had decided to write a slightly too strong assumption for simplicity and readability, but we are willing to change the assumption to a weaker but more complicated one if the reviewers believe it to be important.
>
> Regarding novelty, we argue that this work is more innovative than it may initially appear. In fact, identifying causal effects from CDMGs is significantly more challenging than doing so from acyclic cluster graphs (a topic that has already been explored in a paper published at a top AI conference (Anand et al., 2023) where they showed that the do-calculus as presented by Pearl as applicable as is if the cluster DAG is acyclic). The increased complexity in our setting (when the cluster graph is cyclic) is highlighted by the specific assumption required for identification. The formulation and justification of this assumption is precisely what underpins the novelty of our contribution. Without an additional assumption, the do-calculus as presented by Pearl does not apply to cyclic cluster graphs. We discuss this in Section 4.
>
> The typos mentioned by the reviewer have been duly noted and will be fixed.
> Moreover, we will thoroughly proof read the manuscript to make sure no typo remain in the final version.
>
> - Concerning Definition 5, $\mathbb{X}$ and $\mathbb{Y}$ are sets of micro-variables i.e., variables in the ADMG $\mathcal{G}$. Identifiability is defined for any micro causal effect, but in this paper we are focusing on identifying macro causal effects and thus the we only use Definition 5 for cases where $\mathbb{X}$ and $\mathbb{Y}$ correspond to sets of clusters.
>
> - Concerning the positivity assumptions: We follow (Pearl 2009, p77) for the definition of identifiability and as such we indeed assume that probability is positive everywhere.
> This corresponds to the strict positivity in (On Positivity Condition for Causal Inference - Hwang et al. - 2024).
> It is common in the causal reasoning literature to make such an assumption.
> However, this assumption is too strong and the precise required assumption will depend on the exact graph, causal effect of interest and sequence of do-calculus used to identify.
> There exists some work such as (Hwang et al. 2024) which focus on reducing this assumption but this is beyond the scope of this paper.
> We will add a sentence in the introduction and cite (Hwang et al.2024) to ensure that this assumption is clear to the reader.
>
> - Concerning the definition of Hedge: Putting the bar over or under $\mathbb{X}^*$ does not change the definition, however we will modify this to stay consistent with (Shpitser and Pearl).
>
> Again, we thank the reviewer for their interesting comments, and hope we answered their questions properly. Most importantly, we hope that we have addressed the concerns related to novelty.
> We are willing and ready to respond to further concerns should there be any.

---

> > ### Comment · Reviewer_QfrF · 2025-06-05
> >
> > Thank you for your detailed reply, I have no further questions/comments!

---

### Review · Reviewer_UW5z · 2025-05-17

**Summary Of Contributions:**

The work explores causal-inference identification challenges in a form of partially-defined causal graph, which the authors call “C-DMG” or “Cluster-Directed Mixed Graph”, in manner similar to the work by Anand, et al, 2023[^1] on “cluster-DAGs”. Essentially C-DMGs are a coarse version of an acyclic directed mixed graph model where each node accounts for one or more of “micro-variables”. Such models allow for causal inference to occur even when we don’t have the complete causal model.


Cluster-DAGs put specific structure to how this coarse model can look like, disallowing partitions of the micro-variables that would lead to cycles at the coarse (or cluster) level. C-DMGs do not have this restriction, allowing cycles at the cluster level, and thus require their own analysis. The authors show that standard tools of causal inference at the micro level, that is, d-separation, do-calculus, can be lifted into C-DMGs and they are sound and complete for use there. The authors also show that “hedges”, particular structures that can appear in graphical models, do not extend to C-DMGs, but a modification of them can.

[^1]: |Anand, T.V., Ribeiro, A.H., Tian, J. and Bareinboim, E., 2023, June. Causal effect identification in cluster dags. In _Proceedings of the AAAI Conference on Artificial Intelligence_ (Vol. 37, No. 10, pp. 12172-12179).|
	|Vancouver||

**Audience:**

Yes

**Claims And Evidence:**

Yes

**Requested Changes:**

## Necessary for recommendation

There’s a fair amount of typos in the text. Mentioning some of those here for the convenience of the authors. Please proofread carefully.

### Typos

- In Definition 8, the authors write “$\forall V_1^*, V_n^* \in \mathbb{V}^*_C, \exists V_1^*,\ldots, V_n^*$”. Are the quantifiers correct in this? Perhaps the authors meant that “$\exists$” should apply to the rest of the nodes apart from $V_1, V_n$?

- “it **it** is also readily extendable to …”, please delete the second “it”.
- I think the right article is “**a** hedge”, but both “a” and “an” are being used in the text.
- Page 2, end of page: Please add “-” between the words of “micro” and variables (to make it consistent with how it is later written in the text).
- Page 3, “is at most in two function**s** in \mathbb{F}” (plural form for “function”)
- “an C-DMG” –> “a C-DMG”.
- Page 10, Conclusion, “might exist**s**” (please delete the “s” from "exists").


## Not necessary for recommendation

- Please consider whether a resubmission with a longer format would be appropriate.
- Although C-DMGs are motivated as more general than cluster-DAGs when it comes to structural constraints, it would perhaps help the readers to have some example of where an acyclic Cluster-DAG is not sufficient and a more general C-DMG would be needed. This would give a bit of practical grounding.

And some stylistic suggestions:

- Page 6, top of page, “$\exists 1<i<n$” would be more clear as “$\exists i: 1<i<n$”.

**Strengths And Weaknesses:**

## Strengths

- I can see the usefulness of a partially-identified causal model (as we almost never have the complete ADMG in realistic scenarios) and so studying identification in the case of partial models, like the authors are doing, is bound to be interesting to readers.
- Good sectioning, covering each of the contributions discussed in the intro through separate theorems in each section.
- Multiple examples of toy graphical models provide good intuition for the results.


## Weaknesses

The main weakness is the format; this is an interesting theoretical work that spends a lot of time setting up definitions and stating theorems, with all proofs delegated to the appendix (similarly to the work of Anand, et al.[^1]). While the paper is appropriately sectioned, it’s possible that it would benefit from using the longer TMLR format so that there is space to talk about the proofs a bit in the main text (even if only sketches of the proofs are provided in the main text) and that appropriate time is allocated to review them in more detail. This would also make the paper seem a bit less dense.

[^1]: |Anand, T.V., Ribeiro, A.H., Tian, J. and Bareinboim, E., 2023, June. Causal effect identification in cluster dags. In _Proceedings of the AAAI Conference on Artificial Intelligence_ (Vol. 37, No. 10, pp. 12172-12179).|
	|Vancouver||

---

> ### Author Response · Authors · 2025-05-23
>
> We are grateful to the reviewer for their careful assessment of our work and for providing insightful comments and constructive suggestions.
> In addition, we would like to thank the reviewer for understanding the relevance of working with partially specified graphs as we truly think that this kind of research is bound to become useful in applications.
>
> Regarding the possibility of using more pages give the proof - or at least sketches of them - and discuss the intuitions:
> We had hoped that the examples already given would be sufficient to give convey the main intuitions.
> Moreover, while we can understand the value in adding a few sentences to discuss the intuitions we do not believe that adding proofs or sketches of proofs would be beneficial.
> We believe it would rather complicate the reading of the paper and delay the actual results and messages.
> However, we may be wrong and if the reviewer insists and/or other reviewers agree, we are willing do as recommended.
>
> The typos mentioned by the reviewer have been duly noted and will be fixed.
> Notably, the reviewer is correct concerning the quantifiers in Definition 8,
> $\exists V_1,\cdots,V_{n}$ will be modified to
> $\exists V_2,\cdots,V_{n-1}$.
> The stylistic suggestion will be taken into account.
> Moreover, we will thoroughly proof read the manuscript to make sure no typo remain in the final version.
>
> Again, we thank the reviewer for their insights, and hope we answered their questions thoroughly.
> We are willing and ready to respond to further concerns should there be any.

---

> > ### Comment · Reviewer_UW5z · 2025-05-26
> > **Reply to authors**
> >
> > >Regarding the possibility of using more pages give the proof - or at least sketches of them - and discuss the intuitions: We had hoped that the examples already given would be sufficient to give convey the main intuitions. Moreover, while we can understand the value in adding a few sentences to discuss the intuitions we do not believe that adding proofs or sketches of proofs would be beneficial. We believe it would rather complicate the reading of the paper and delay the actual results and messages. However, we may be wrong and if the reviewer insists and/or other reviewers agree, we are willing do as recommended.
> >
> > Thank you. After rereading the work, I think the format won't be a blocker for acceptance.
> >
> > I have no further questions.

---

### Review · Reviewer_p9DU · 2025-05-18

**Summary Of Contributions:**

This paper tackles the problem of identifying causal effects when in cluster-directed mixed graphs (C-DMGs) where certain vertices in an acyclic directed mixed graphs (ADMGs) might be grouped together. The authors showed that d-separation and do-calculus remain sound and complete for identifying macro causal effects in general C-DMGs over ADMGs when the size of clusters is either unknown or when each cluster has at least two variables. Under the same assumptions, they also showed that existence of a SC-hedge (an extension of hedge introduced by prior work) implies non-identifiability in general C-DMGs.

**Audience:**

Yes

**Broader Impact Concerns:**

NIL

**Claims And Evidence:**

Yes

**Requested Changes:**

Requested changes:
- Regarding the weakness comment about Assumption 1, can you be more explicit about it? I understand that you gave an example in Section 4 but could you add some prose to explicitly explain when and why Assumption 1 is necessary for your results to hold? Which part of your proofs explicitly relies on this assumption?
- See the weakness comment about completeness of SC-hedge

Minor typos:
- Third bullet on Page 2: "SC-hedge Ferreira & Assaad (2025)" should "SC-hedge (Ferreira & Assaad, 2025)". Probably some a citep typo.
- Figure 1(a) caption: Missing space between "ADMG" and "1"
- Theorem 3, Rule 2: c^W shoudl be c_W
- First paragraph on Page 8 starting with "We first focus on...": The sentences that follow have a lot of typographic errors. In your notation for conditional independence with respect to mutilated graphs, the c's in the brackets should be capitalized while the c's in the subscript should be calligraphic?

**Strengths And Weaknesses:**

# Strengths

- The paper is full of illustrative examples to give intuition behind the results and discussions.
- The authors were upfront about the limitations of their results and discussed them appropriately.

# Weaknesses

- Three limitations have been identified by the authors in the conclusions, and I feel that it has been discussed adequately.
- It is not clear to me when when why Assumption 1 is necessary for the results to hold. The only time I see it referenced in the proofs is for Theorem 5. If I have missed it, please point me to the discussion. Thank you.
- If I recall correctly, hedges were sound and complete for ADMGs. If I did not misread the paper, the authors showed that soundness of SC-hedges, i.e., "under Assumption 1, existence of SC-hedge implies non-identifiability". Is there a corresponding completeness result, i.e. "under Assumption 1, non-identifiability implies existence of SC-hedge"? If it is not possible, it should be discussed why it is not complete for C-DMGS. If I have missed it, please point me to the discussion. Thank you.

---

> ### Author Response · Authors · 2025-05-23
>
> We sincerely thank the reviewer for their thorough evaluation of our paper and for the thoughtful comments and valuable suggestions.
>
> Why is Assumption 1 necessary?
> - Assumption 1 allows to know that if a directed path $\langle C_1 \rightarrow \cdots \rightarrow C_n\rangle$ exists in the C-DMG then there exists an ADMG in which the directed path $\langle V_1 \rightarrow \cdots \rightarrow V_n \rangle$ exists with $\forall i,~V_i\in C_i$. Indeed, in Figure 5, in which Assumption 1 is not verified, there is no compatible ADMG with a directed path from $Y_1$ to $X_1$. If Assumption 1 is violated, Theorems 1 and 3 still hold but
> Theorems 2, 4 and 5 are no longer true. We use Assumption 1 in the proof of Theorem 2 and 5 and we use Theorem 2 in the proof of Theorem 4, however we agree that this could be stated more clearly and we are willing to do so in the finalized version.
>
> Is the SC-hedge criterion complete?
> -
> It is true that Theorem 5 only states the soundness of the SC-hedge criterion and does not mention anything regarding its completeness.
> Until now we have not been able to prove the completeness of the SC-hedge criterion even though we could not find a counter example that could prove its non-completeness.
> The proof of the completeness of the hedge criterion in ADMGs relies on the completeness of the ID algorithm which cannot be used in cyclic graphs (e.g., C-DMGs).
> Therefore, proving the completeness of the SC-hedge criterion is left for future work.
> We are willing to add a few words to clarify this point in the final version, unfortunately we will not be able to state the completeness of the SC-hedge criterion.
>
> The typos mentioned by the reviewer have been duly noted and will be fixed.
> Moreover, we will thoroughly proof read the manuscript to make sure no typo remain in the final version.
>
> Again, we thank the reviewer for their interesting comments, and hope we answered their questions properly.
> We are willing and ready to respond to further concerns should there be any.

---

> > ### Comment · Reviewer_p9DU · 2025-05-28
> >
> > Thank you very much for the clarifications! It would be helpful if you add in these discussions into the revision. I have no further concerns.

---

### Decision · Action_Editor_8Kw3 · 2025-06-13

**Recommendation:** Accept with minor revision

**Additional Comments:**

Following the discussion phase and the authors' clarifications, the reviewers recommended acceptance. Their feedback reflects the perceived relevance of the problem setting and the overall clarity of the theoretical contributions.

Requested improvements are related to presentation and clarification:

- Clearly explain where Assumption 1 is invoked in the proofs and why it is necessary.
- Emphasize that the SC-hedge criterion is sound but its completeness remains open, clarify this as a limitation.
- Fix remaining typographical issues.

**Audience:**

Yes

**Audience Explanation:**

This work is well-aligned with the TMLR readership. It targets a fundamental yet underexplored challenge in causal inference: reasoning over partially specified graphs. The topic is timely and relevant for researchers working on **identifiability**, **graphical models**, and **causal discovery**, especially in fields like epidemiology or biology where coarse or uncertain structural knowledge is the norm.

While some reviewers viewed the novelty as moderate, given that the main results are extensions of known tools, the generalization to C-DMGs, is **non-trivial and well-motivated**. The paper is mathematically dense but conceptually clear, and the examples aid accessibility. Overall, the work makes a solid contribution to the causal inference literature and will be particularly relevant to researchers interested in theoretical aspects of identifiability in partially specified models.

**Claims And Evidence:**

Yes

**Claims Explanation:**

The paper presents a well-executed theoretical contribution in the area of causal inference under partial structure specification. It tackles the problem of identifying **macro causal effects** in **cluster-directed mixed graphs (C-DMGs)**, a generalization of ADMGs that permits cycles and variable groupings into clusters. The authors prove that under a mild assumption on cluster sizes, both **d-separation** and the **do-calculus** are sound and complete for identifying such effects. They also introduce **SC-hedges** as a graphical condition for non-identifiability.

The reviewers generally found the work **technically rigorous**, well-structured, and convincingly argued. The theoretical results are supported by clean examples and motivated by practical scenarios where full causal graphs are unavailable. While the work builds on existing concepts, it extends them to a setting with significant new challenges and relevance.

The authors addressed key questions raised during the discussion phase, including the role of Assumption 1 and the scope of the SC-hedge criterion. Following their clarifications, the reviewers confirmed that their concerns had been resolved.